# DriveE2E: Closed-Loop Benchmark for End-to-End Autonomous Driving through Real-to-Simulation

## Abstract

Closed-loop evaluation is increasingly critical for end-to-end autonomous driving. Current closed-loop benchmarks using the CARLA simulator rely on manually configured traffic scenarios, which can diverge from real-world conditions, limiting their ability to reflect actual driving performance. To address these limitations, we introduce a simple yet challenging closed-loop evaluation framework that closely integrates real-world driving scenarios into the CARLA simulator with infrastructure cooperation. Our approach involves extracting 800 dynamic traffic scenarios selected from a comprehensive 100-hour video dataset captured by high-mounted infrastructure sensors, and creating static digital twin assets for 15 real-world intersections with consistent visual appearance. These digital twins accurately replicate the traffic and environmental characteristics of their real-world counterparts, enabling more realistic simulations in CARLA. This evaluation is challenging due to the diversity of driving behaviors, locations, weather conditions, and times of day at complex urban intersections. In addition, we provide a comprehensive closed-loop benchmark for evaluating end-to-end autonomous driving models. Code and dataset examples are in the supplementary materials.

## 1 Introduction

End-to-End Autonomous Driving (E2EAD) has shown great advances and potential. Effective evaluation is essential for assessing the driving capabilities of E2EAD models, thereby advancing research and promoting the development of improved algorithms. Traditionally, E2EAD performance has been assessed using open-loop evaluation, which operates on pre-recorded expert driving trajectories and corresponding sensor data, as seen in datasets such as nuScenes Caesar et al. (2020). In this setting, the model passively predicts actions without influencing future observations, making the task resemble trajectory prediction Zhai et al. (2023); Li et al. (2024b). As a result, open-loop

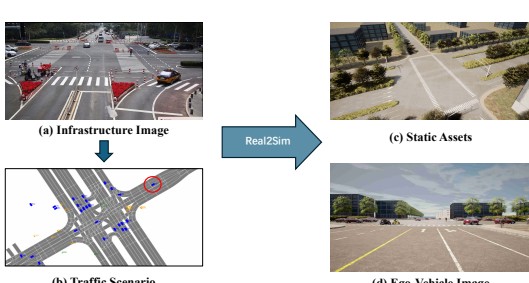

Figure 1: **Real2Sim: From Real-World Infrastructure to Simulated Vehicle.** (a) Images captured by multi-view infrastructure cameras, mounted at elevated positions to provide a broader field of view than vehicle-mounted sensors. (b) Traffic participants extracted from infrastructure images, effectively reducing occlusion. Red circle denotes the selected ego vehicle. (c) Static digital-twin intersection assets, with appearance aligned to the original scene. (d) Simulated image generated.

evaluation provides limited insight into vehicle-environment interactions and real-time decision-making. In contrast, closed-loop evaluation continuously updates observations based on the ego vehicle's actions, allowing the E2EAD model to control the vehicle using its own decisions. This interaction-rich setting offers a more realistic and rigorous assessment of model performance.

Closed-loop evaluation is currently performed primarily in simulators, due to the high cost of on-road testing and the lack of a reliable world model. Among available platforms, CARLA Dosovitskiy et al. (2017) has emerged as the most widely adopted simulator in the autonomous driving community, owing to its powerful rendering engine and operational efficiency. Notable benchmarks such as Carla LB V2 Contributors (2024) and Bench2Drive Jia et al. (2024) are both built upon the CARLA. However, these benchmarks typically rely on manually constructed driving scenarios, in

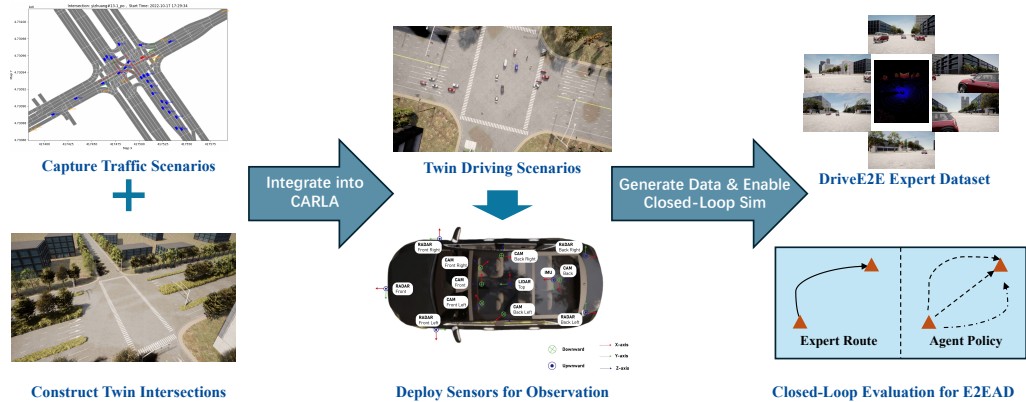

Figure 2: **Overview of DriveE2E.** We begin by capturing traffic scenarios from infrastructure sensor data and constructing corresponding digital twins of real-world intersections. These elements are then loaded into CARLA to create twin driving scenarios, with sensors equipped on the designated ego autonomous vehicle, adopting the nuScenes Caesar et al. (2020) configuration. Along the expert-defined route, we collect expert data for training E2EAD models. Using the planning output from the E2EAD systems, we evaluate their driving performance in a closed-loop manner.

which traffic participants, their behaviors, and environmental conditions are manually configured. While effective for testing within simulation, this approach can lead to a research-development gap, as it fails to reflect the complexities and patterns of real-world traffic. As a result, E2EAD research based on these benchmarks will be limited in practical relevance and slowing innovation toward deployable solutions. To better align the E2EAD research community with the on-road testing and the practical industry needs, the Real2Sim (Real-to-Simulation) introduces real-world elements into the simulator, providing a promising path to enhance CARLA's realism.

In this work, we introduce DriveE2E, a closed-loop E2EAD benchmark built on the CARLA simulator using a Real2Sim approach with offline infrastructure cooperation. As illustrated in Figure 1, real-world elements—including dynamic traffic scenarios and high-fidelity static road environment assets—are imported into CARLA to enable more realistic simulation. DriveE2E features two key characteristics: **1) Offline infrastructure cooperation.** Existing Real2Sim methods typically extract traffic scenarios—including vehicles and pedestrians—from single-vehicle autonomous driving data. However, such data often suffer from significant information loss due to limited perceptual range and occlusions, particularly when relying on cost-effective camera sensors. To address this, DriveE2E leverages infrastructure-mounted sensors, which offer elevated viewpoints and broader coverage, to extract traffic elements more comprehensively. These elements are then imported into the simulator, enabling the ego vehicle to perceive necessary traffic participants from any novel perspective. We refer to this use of infrastructure data as offline mode, where the data is not accessed in real time to directly enhance driving performance Yu et al. (2024); Cui et al. (2022); Hao et al. (2025b). **2) High-fidelity digital twin.** We reconstruct key road elements—including road topology, lane geometry, and surrounding buildings—to create a digital twin of the original driving environment. Additionally, we capture and integrate critical environmental information such as weather conditions and lighting, ensuring that the simulation environment closely mirrors real-world conditions. This enhances the fidelity of DriveE2E, better aligned with real-world driving scenarios.

Specifically, we constructed high-fidelity digital twins of 15 urban intersections, each featuring diverse road layouts and topologies to reflect a broad spectrum of real-world traffic conditions. From over 100 hours of multi-view footage captured by infrastructure-mounted cameras at these intersections, we selected 800 representative clips to generate traffic scenarios encompassing eight distinct driving behaviors, six weather conditions, and various times of day—from morning to night. The extracted traffic elements and constructed intersection assets were then imported into the corresponding digital twins, forming the basis of our **Twin Driving Scenarios**. To construct the benchmark, we designated an ego vehicle in each clip and collected 800 sensor data sequences along the original routes in DriveE2E's twinned scenarios as expert demonstrations. Five classical E2EAD models were then trained via imitation learning and evaluated in a closed-loop setting within the DriveE2E

Table 1: Comparison with related closed-loop evaluation benchmarks based on simulators. Unlike CARLA, Panda3D Goslin and Mine (2004) produces sensor data with much lower fidelity. MetaDrive Li et al. (2022a), for example, only considers geometry mapping but lacks appearance consistency in traffic participants, road appearance, and surroundings. 'Inf.' denotes infrastructure.

| Benchmark | Simulator | Fidelity | Real2Sim | Real Source | Consistency | Expert |
|---|---|---|---|---|---|---|
| Longest6 Chitta et al. (2023) | CARLA | Medium | ✗ | - | - | ✓ |
| Safebench Xu et al. (2022) | CARLA | Medium | ✗ | - | - | ✗ |
| CARLA LB V2 Contributors (2024) | CARLA | Medium | ✗ | - | - | ✗ |
| Bench2Drive Jia et al. (2024) | CARLA | Medium | ✗ | - | - | ✓ |
| MetaDrive Li et al. (2022a) | Panda3D | Low | ✓ | Vehicle View | Low | ✗ |
| ScenarioNet Li et al. (2023) | Panda3D | Low | ✓ | Vehicle View | Low | ✗ |
| **DriveE2E (Ours)** | CARLA | Medium | ✓ | Inf. View | High | ✓ |

simulator. The whole process is shown in Figure 2. It is important to note that we currently use a simplified closed-loop protocol—log-replay mode—where non-ego traffic participants strictly follow their recorded trajectories without responding to the ego-vehicle actions. This choice is motivated by both efficiency and reliability: DriveE2E scenes involve many traffic agents, and introducing reactive behavior for all of them would make large-scale evaluation impractical. Moreover, existing reactive models remain immature, as they often imitate trajectories rather than relying on agents' own observations, can produce unstable behaviors in dense intersections, and lack robust criteria to ensure realistic interactions. To avoid these issues and preserve the fidelity of real-world behaviors, we adopt the log-replay approach. **Notably, DriveE2E is the first CARLA-based, closed-loop benchmark for end-to-end autonomous driving grounded in a Real2Sim approach, specifically designed to narrow the gap between simulation-based evaluations and real-world testing.**

Our contributions can be summarized as follows.

- We propose an infrastructure-view-enhanced real-to-simulation framework for closed-loop evaluation of E2EAD, which integrates real-world traffic elements and twin intersection assets into the CARLA simulator through infrastructure-based sensing. This approach enhances the realism of simulation environments and aligns model evaluation more closely with real-world testing needs.

- We construct high-fidelity digital twins of 15 urban intersections and select 800 real-world traffic scenarios from more than 100 hours of infrastructure sensor data. These scenarios capture diverse driving behaviors, geographic locations, weather conditions, and times of day, while faithfully replicating road geometry and environmental structures from the original scenes.

- We establish a comprehensive closed-loop benchmark for E2EAD by evaluating several baselines, including UniAD Hu et al. (2023b), VAD Jiang et al. (2023), TCP Wu et al. (2022), AD-MLP Zhai et al. (2023), and MomAD Song et al. (2025). Furthermore, we provide an expert dataset derived from the digital twin scenarios to support imitation learning-based E2EAD training.

## 2 RELATED WORK

**End-to-End (E2E) Autonomous Driving.** End-to-end (E2E) approaches integrate perception, prediction, and planning into a single, differentiable model Hu et al. (2023b); Chen et al. (2024b); Chib and Singh (2024); Hao et al. (2025a), optimizing the system holistically by transforming raw sensor data directly into driving actions Jia et al. (2023); Shao et al. (2024). To acquire driving skills, some approaches Codevilla et al. (2018); Prakash et al. (2021); Wu et al. (2022) leverage imitation learning (IL), where models learn from expert demonstrations. In contrast, others Liang et al. (2018); Kendall et al. (2019); Jia et al. (2023) utilize reinforcement learning (RL), iteratively learning by interacting with the environment. Imitation learning, compared to reinforcement learning, shows promise for E2E systems in harnessing large-scale datasets effectively. Recent advancements in E2E systems focus on transformer-based models Prakash et al. (2021); Chitta et al. (2023); Shao et al. (2023a); Jaeger et al. (2023); Shao et al. (2023b), LLM-enhanced models Pan et al. (2024); Chen et al. (2024a); Xu et al. (2024); Fu et al. (2024); Sima et al. (2024), and world models Zheng et al. (2024); Li et al. (2024a); Wang et al. (2023). These advancements tackle key

challenges—such as generalization—and lead to improved performance, significantly accelerating progress in autonomous driving.

**Evaluation Benchmarks for E2EAD.** Benchmarks play a crucial role as they provide standardized metrics for measuring progress and help assess the practical applicability of E2EAD systems. There are two primary methods for evaluating E2EAD algorithms. The first is open-loop evaluation Caesar et al. (2020); Dauner et al. (2024), which has been widely adopted in E2EAD assessments Hu et al. (2023b); Jiang et al. (2023). However, by restricting the ego vehicle's observations to route-specific states, it limits assessment of long-horizon planning in E2EAD models Zhai et al. (2023); Li et al. (2024b). The second is closed-loop evaluation, which typically relies on simulators to enable interaction between the ego vehicle and environmental agents. The most prominent end-to-end closed-loop simulator is open-source CARLA Dosovitskiy et al. (2017) with its realistic rendering and high efficiency. CARLA Dosovitskiy et al. (2017) has spawned several benchmarks such as CARLA LB V2 Contributors (2024), Longest6 Chitta et al. (2023), V2XVerse Liu et al. (2024), and Bench2Drive Jia et al. (2024). However, these benchmarks rely on artificially created scenarios rather than real-world trajectories, which may misrepresent real-world testing needs. An emerging direction is to leverage generative methods, such as diffusion-based Yang et al. (2024), GPT-based Hu et al. (2023a), NeRF-based Tonderski et al. (2024), and 3DGS-based Cao et al. (2025) approaches, to generate realistic images for closed-loop evaluation. However, these novel-view observation synthesis remain insufficient for closed-loop evaluation needs.

The Real2Sim (Real-to-Simulation) approach has shown great potential to bridge the evaluation gap between simulation and real-world testing. Several works, such as MetaDrive Li et al. (2022a) and ScenarioNet Li et al. (2023), attempt to load real-world datasets like nuScenes Caesar et al. (2020) into simulators such as Panda3D Goslin and Mine (2004), which shows low rendering fidelity compared to CARLA Dosovitskiy et al. (2017). However, these efforts typically focus only on importing geometric elements while ignoring visual appearance, and they rely solely on vehicle-view data for agent extraction. This results in limited consistency with the original real-world environment. Furthermore, they do not provide standardized implementations of classical E2EAD models for benchmarking. In contrast, DriveE2E builds high-fidelity static assets and incorporates traffic scenarios from an infrastructure-view perspective, enabling more realistic and consistent closed-loop evaluations of E2EAD methods. A comparison with related benchmarks is presented in Table 1.

## 3 DriveE2E

To advance end-to-end autonomous driving research in a direction more aligned with real-world needs, we introduce DriveE2E, a closed-loop benchmark built on the CARLA simulator, specifically designed for evaluating E2EAD systems using a Real2Sim approach and offline infrastructure cooperation. An overview of the DriveE2E framework is presented in Figure 2. The benchmark pipeline consists of the following key components: dynamic traffic scenario acquisition from infrastructure sensor data (Sec.3.1); static intersection asset construction (Sec.3.2); ego vehicle assignment and sensor configuration (Sec. 3.3); and integration of dynamic scenarios with their corresponding digital-twin intersections into the CARLA simulator, including visual-appearance configuration. We also detail the expert data collection process for imitation learning-based training (Sec.3.4) and describe the closed-loop evaluation procedure for end-to-end models within DriveE2E (Sec.3.5). Finally, we provide dataset statistics in Sec. 3.6, and simulation-to-real comparison in Appendix.

### 3.1 Dynamic Traffic Scenario Acquisition

**Equipment.** Infrastructure-mounted sensors, installed at elevated positions, offer broader perception capabilities Yu et al. (2022); Hao et al. (2024). We selected 15 intersections from the Beijing High-level Autonomous Driving Demonstration Zone, as shown in Figure 3(a). At each intersection, four pairs of roadside cameras, along with additional blind-spot cameras, were installed at elevated heights to ensure full coverage of the intersection area, as illustrated in Figure 3(b). All cameras were precisely calibrated to support accurate 3D perception. In addition, the equipment was configured to capture real-time traffic light signals.

**Data Collection and Annotation.** We collected sensor sequence data over a 100-hour period, along with recording traffic light signals at the same frequency. In addition, we obtained weather and illumination from a weather service. The collected sensor data was then processed using trained 3D object detection Rukhovich et al. (2022) and tracking models Weng et al. (2020) combined with multiview fusion, generating trajectory sequences with 3D bounding box information. Each box was assigned a class label from 8 categories and a unique trajectory ID. This multi-view perception provided comprehensive coverage of almost all traffic participants at each intersection. Each trajectory was also assigned a quality score based on completeness and compliance with the traffic rules. Scenarios with low scores were discarded, ensuring a high-quality traffic scenario database. Finally, we select 800 scenarios to encompass a wide range of scenes and driving conditions.

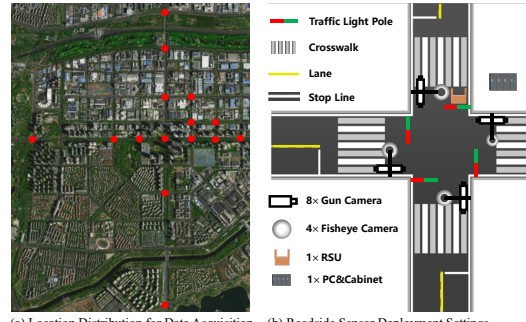

(a) Location Distribution for Data Acquisition   (b) Roadside Sensor Deployment Settings

Figure 3: Dynamic Traffic Scenario Acquisition.

## 3.2 STATIC INTERSECTION ASSET CONSTRUCTION

We first obtained HD maps covering the 15 selected intersections, organized in a format similar to Argoverse Chang et al. (2019). These HD maps include vectorized representations of the centerlines of the lane, crosswalks, and stop lines. The maps were then imported into RoadRunner MathWorks (2023) [1], where we carefully refined and corrected elements of the structure of the road by referencing high-resolution satellite images and street-view images to ensure geometric accuracy.

To enhance realism, we also incorporated surrounding elements such as buildings using data from OpenStreetMap OSM contributors (2023) [2]. We configured the appearance attributes for these elements to match the visual appearance of real-world scenes. Additionally, we manually added traffic lights and the corresponding light poles to replicate the actual infrastructure of the intersection. All road structures, environmental elements, and traffic lights were then integrated and aligned in Blender Blender Studio (2023), for fine-grained manual adjustments to ensure spatial consistency.

Finally, these components were unified into a single simulation environment for each intersection, forming a set of high-fidelity static digital twin assets. These twins closely mirror the original intersections while conforming to CARLA's formatting requirements. The complete generation process is illustrated in the Appendix. This complex pipeline produces digital assets that preserve the structural and visual fidelity necessary for realistic autonomous driving research.

## 3.3 EGO VEHICLE ASSIGNMENT AND SENSOR CONFIGURATION

DriveE2E is designed to evaluate the performance of single-vehicle autonomous driving systems, which requires explicitly selecting a vehicle for evaluation. In contrast to approaches that import traffic scenarios from the vehicle's perspective—where the data collection vehicle is by default treated as the autonomous test vehicle—DriveE2E requires explicitly designating a target vehicle as the ego vehicle in each scenario.

Table 2: Key Sensor Specifications for Ego Vehicle.

| Sensor | Details |
|---|---|
| 1x LiDAR | 64 channels, 85-meter range, $360°$ horizontal FOV, $+10°$ to $-30°$ vertical FOV |
| 6x Camera | Surround coverage, RGB, 900x1600 resolution, JPEG compressed |
| 5x Radar | 100-meter range |
| 1x IMU&GPS | Position, heading, speed, acceleration, and angular velocity |

Specifically, we analyzed the driving behaviors of all vehicles in each scene and selected a candidate vehicle that remained fully visible throughout the clip. The final ego vehicle was chosen based on diversity and representativeness of its driving behaviors. Driving behaviors distribu-

---

[1] RoadRunner MathWorks (2023): a 3D environment editing tool used to design and edit traffic and road scenes for the simulation and testing of autonomous driving systems.

[2] OpenStreetMap OSM contributors (2023): a global, user-contributed, open-source map database.

tion is provided in Section 3.6. This designated ego vehicle was equipped with a standard sensor suite—including LiDAR, cameras, radars, IMU, and GPS—following a configuration similar to that used in nuScenes Caesar et al. (2020). Sensor configuration is provided in Table 2 and Figure 2.

## 3.4 EXPERT DATASET COLLECTION

Current end-to-end autonomous driving models are typically trained via imitation learning using expert demonstrations. To ensure that the DriveE2E benchmark can be fairly and effectively adopted by the research community, we also release the dataset collected within the DriveE2E environment.

Specifically, we first load the twinned driving scenarios into the CARLA simulator. For each of the 800 selected scenes, we import the corresponding static intersection assets and instantiate all traffic participant actors—excluding the designated ego vehicle—into the simulation. These actors are mapped to CARLA blueprints based on their 3D attributes such as category and size. Once the environment is initialized, we drive the assigned ego vehicle, equipped with the sensor suite, along its original real-world trajectory as defined in the dynamic scenario acquisition process.

Sensor data is recorded at 10 Hz. The collected dataset includes LiDAR point clouds, multi-view RGB images, radar points, GPS trajectories, and top-view images. In addition to raw sensor data, we provide 3D bounding box annotations—adjusted to account for potential discrepancies caused by limitations in CARLA's blueprint assets, which may not perfectly match the real-world object dimensions—and ego vehicle state information. Both are essential for training and evaluating autonomous driving models. Moreover, we export HD maps of the 15 selected intersections from the DriveE2E simulator. This full collection constitutes the **DriveE2E Expert Dataset**.

## 3.5 CLOSED-LOOP EVALUATION

Closed-loop evaluation for end-to-end autonomous driving (E2EAD) enables an autonomous vehicle to interact with its surrounding traffic environment and respond to dynamic changes in real time. This approach continuously updates the observed environment based on the vehicle's actions, allowing for a more comprehensive assessment of its decision-making capabilities.

In DriveE2E, the autonomous ego vehicle is tasked with navigating from a source location $(x_{src}, y_{src})$ to a destination $(x_{dst}, y_{dst})$ within a given scenario, where both points lie on its original real-world trajectory. The E2EAD system receives raw sensor data (e.g., multi-view images), GPS coordinates, and a set of downsampled waypoints from the original ego route as inputs. The model then outputs either low-level control commands (steering angle, throttle, brake) or high-level future waypoints, which are subsequently translated into control commands by the CARLA simulator.

Ideally, other traffic participants should also react to the ego vehicle's actions during closed-loop evaluation. In our current implementation, we adopt the simplest form—log-replay mode—in which the ego vehicle is controlled by the E2EAD model, while all other agents follow their original recorded trajectories without responding to the ego's actions. We acknowledge that this mode has limitations in evaluating realistic interactions. We provided a detailed comparison across DriveE2E, Open-loop and Closed-loop evaluations in Appendix VI. Future versions of DriveE2E aim to incorporate more interactive agent behaviors, such as rule-based models (e.g., the Intelligent Driver Model (IDM) Treiber et al. (2000)) or learned reactive policies, to enhance the realism and fidelity of closed-loop evaluation.

**Evaluation Metrics.** Here we adopt two metrics to evaluate the performance of the E2EAD system, following CARLA LB V2 Contributors (2024) and Bench2Drive Jia et al. (2024): Success Rate (SR) and Driving Score (DS). Detailed explanations are provided in Appendix.

## 3.6 DATA ANALYSIS

DriveE2E includes 800 twinned driving scenarios located across 15 urban intersections, capturing a wide range of driving behaviors, weather conditions, and times of day—from morning to night.

**Static Twin Intersections Assets.** The 15 selected intersection assets feature diverse and complex road elements and topological structures, enabling comprehensive evaluation of an E2EAD

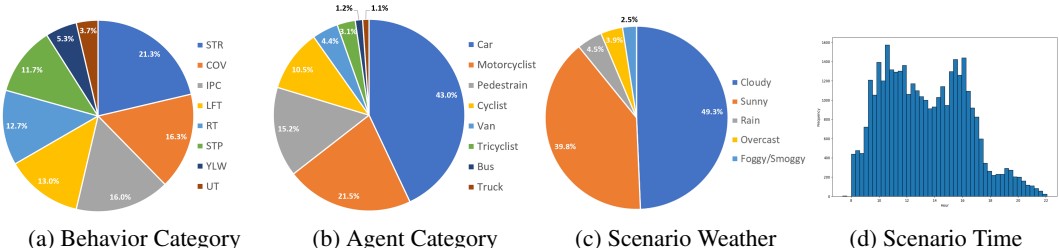

Figure 4: Data distribution of the driving scenarios.

system's road understanding capabilities. Each intersection includes realistic components such as signage, lane markings, crosswalks, stop lines, traffic light poles, traffic lights, and adjacent buildings—together contributing to a simulation environment that closely mirrors real-world complexity. Visualizations of all intersection assets are provided in the Appendix.

**Driving Behaviors.** DriveE2E identifies and categorizes 8 typical scenario types at intersections from 800 real-world traffic scenarios. These behaviors include Interaction with Pedestrians and Cyclists (IPC), Competing with Other Vehicles (COV), Passing through during Yellow Lights (YLW), Making a U-turn (UT), Stopping at Red Lights (STP), Going Straight through Intersection (STR), Making a Left Turn (LFT), and Making a Right Turn (RT). The distribution of these behaviors is illustrated in Figure4(a). These eight scenarios are further refined into 14 specific sub-scenarios based on turning conditions and anomalies, which are explained in the appendix.

**Traffic Agents.** As shown in Figure 4(b), DriveE2E supports eight types of traffic agents. The majority consist of cars, motorcycles, pedestrians, and cyclists, alongside less frequent agents such as trucks, buses, tricycles, and vans. This diverse agent composition ensures realistic interactions and supports rigorous evaluation of E2EAD models in dense, heterogeneous traffic environments.

**Weather and Light Conditions.** The distributions of weather and time conditions are shown in Figure 4(c–d). DriveE2E includes six types of weather conditions, including rare but challenging scenarios such as rain, overcast, and fog. Temporal diversity is also ensured, with real-world trajectories spanning the full day—from early morning to late night—and covering peak traffic periods, where complex and unpredictable interactions are more likely to arise.

## 4 EXPERIMENTS

### 4.1 BASELINES AND IMPLEMENTATION DETAILS

We implement five classical End-to-End Autonomous Driving (E2EAD) models as baseline methods. The 800 expert data clips were divided into training, validation, and test sets using a 2:1:1 split. Model training was conducted via imitation learning on A100 GPUs using the 400 training clips. Both closed-loop and open-loop evaluations were performed on the validation set. Open-loop performance is reported as L2 Error (m) at 1s and 2s horizons. These models include UniAD Hu et al. (2023b), VAD Jiang et al. (2023), AD-MLP Zhai et al. (2023), TCP Wu et al. (2022), and MomAD Song et al. (2025). Explanations about these baseline models are provided in Appendix.

### 4.2 MAIN RESULTS

**Open-Loop Evaluation Results.** Open-loop evaluation has been widely criticized Zhai et al. (2023); Li et al. (2024b). However, as shown in Table 3, AD-MLP exhibits high L2 error, with an average error reaching 8.36 m, while VAD, UniAD and MomAD all largely outperform AD-MLP (0.89 m, 1.08 m and 0.98 m *vs.* 8.36 m). This result contrasts with the performance observed on nuScenes Zhai et al. (2023), where relying solely on past ego status led to strong planning outcomes. This result raises a crucial question for the autonomous driving community: **Is open-loop evaluation really worthless for end-to-end algorithms?** We believe the discrepancy is understandable, as

Table 3: Open-Loop and Closed-Loop Evaluation Results of Different Baseline Models in DriveE2E. The average time cost for evaluating each model per scene in the closed-loop setting is also reported.

| Methods | L2 Error (m) ↓ | | | Closed-Loop | | |
|---|---|---|---|---|---|---|
| | 1s | 2s | Avg. | SR (%) ↑ | DS (%) ↑ | Test Time (s/scene) |
| AD-MLP Zhai et al. (2023) | 4.98 | 11.75 | 8.36 | 1.0 | 29.01 | 35.11 |
| TCP Wu et al. (2022) | 1.67 | 3.45 | 2.56 | 10.00 | 48.47 | 32.93 |
| TCP-ctr Wu et al. (2022) | - | - | - | 3.00 | 26.73 | 29.55 |
| TCP-traj Wu et al. (2022) | - | - | - | 25.50 | 61.52 | 31.28 |
| VAD Jiang et al. (2023) | 0.62 | 1.16 | **0.89** | 35.00 | 62.29 | 79.91 |
| UniAD Hu et al. (2023b) | 0.69 | 1.47 | 1.08 | **47.00** | **77.62** | 103.06 |
| MomAD Song et al. (2025) | 0.68 | 1.28 | 0.98 | 29.64 | 60.98 | 104.03 |

DriveE2E incorporates a broader range of driving behaviors, unlike nuScenes, where most behaviors are relatively straightforward. Moreover, we contend that open-loop evaluation still holds reference value when the evaluation scenarios are complex enough, further emphasizing the significance of DriveE2E for open-loop evaluation. Additionally, VAD, UniAD and MomAD all significantly outperform AD-MLP and TCP (0.89 m, 1.08 m and 0.98 m *vs.* 8.36 m and 2.56 m), which is expected given the increased challenge of our benchmark and the fact that UniAD and VAD are specifically designed for planning tasks. Furthermore, VAD achieves a lower L2 error and performs better than UniAD and MomAD (0.89 m vs. 1.08 m and 0.98 m) in open-loop evaluation.

**Closed-loop Evaluation Results.** Both AD-MLP and TCP show very low success rates and driving scores, with AD-MLP achieving 1.0 SR and 29.01 DS, and TCP reaching 10.00 SR and 48.47 DS. In contrast, VAD achieves the best performance in open-loop evaluation up to 0.89 Avg. L2 Error, and UniAD achieves the best performance in closed-loop evaluations, with a success rate of 47.00 and a driving score of 77.62. These results suggest that relying solely on past ego states is insufficient for producing effective planning outputs in complex traffic environments. Additionally, we report the evaluation time per scene for each model. UniAD Hu et al. (2023b) requires the longest time at 103.06 s/scene, though this remains within an acceptable range for evaluation.

**Relationship between Closed-loop and Open-loop Evaluation Results.** To some extent, open-loop and closed-loop evaluations are related. For example, AD-MLP, which has the highest L2 error, also exhibits the worst driving performance in closed-loop evaluation. Conversely, VAD and UniAD perform well in both open-loop and closed-loop assessments. This suggests that open-loop evaluations with difficult and diverse driving scenarios can provide insight into driving ability evaluation. However, the results across different methods do not always show a strictly consistent pattern between open-loop and closed-loop evaluations, as shown in the comparison between UniAD and VAD (Table 3). This is because open-loop outputs do not necessarily correlate positively with the outcomes of closed-loop evaluations, which involve some level of interaction. Therefore, closed-loop evaluation remains essential for assessing driving ability.

### 4.3 PERFORMANCE ON DIFFERENT BEHAVIORAL SCENARIOS

We also evaluated four trained E2EAD models across the eight different behavior categories in DriveE2E, with the results presented in Table 4. The performance of E2EAD systems in certain categories, such as IPC and COV, is worse compared to the STP category. This is because scenarios like IPC and COV involve interactions with other traffic participants, such as pedestrians and motor vehicles, which place greater demands on driving ability. In contrast, behaviors like stopping at red lights (STP) are simpler and require relatively lower driving skill.

### 4.4 ABLATION STUDY FOR TRAFFIC PARTICIPANTS MISSING

To investigate the impact of infrastructure-view versus vehicle-view data in Real-to-Simulation closed-loop evaluation for E2EAD models, we design and implement experiments focusing on occlusion-induced missing information in vehicle-view scenarios.

Table 4: Closed-loop Evaluation for Different Behavioral Scenarios.

| Models | Success Rate (%) for Different Behavior Categories ↑ | | | | | | | |
|---|---|---|---|---|---|---|---|---|
| | COV | IPC | UT | YLW | STR | LFT | RT | STP |
| AD-MLP Zhai et al. (2023) | 0.00 | 0.00 | 0.00 | 5.88 | 0.00 | 0.00 | 0.00 | 4.55 |
| TCP Wu et al. (2022) | 16.67 | 2.94 | 40.00 | 5.88 | 2.78 | 3.85 | 12.50 | 22.73 |
| TCP-ctr Wu et al. (2022) | 5.56 | 2.94 | 0.00 | 0.00 | 0.00 | 7.69 | 0.00 | 4.55 |
| TCP-traj Wu et al. (2022) | 25.00 | 28.57 | 20.00 | 40.00 | 21.21 | 8.70 | 14.29 | 88.89 |
| VAD Jiang et al. (2023) | 38.89 | 32.35 | 20.00 | 23.53 | 36.11 | 46.15 | 41.67 | 22.73 |
| UniAD Hu et al. (2023b) | **40.63** | **46.43** | **60.00** | **53.33** | 39.39 | **65.22** | **52.38** | **100.00** |
| MomAD Song et al. (2025) | 19.44 | 23.53 | 20.00 | 47.06 | **41.67** | 42.31 | 20.83 | 18.18 |

**Implementation.** Unlike infrastructure-view data, which captures the full intersection and all traffic participants, vehicle-view sensor data is often limited by occlusions. Here we construct vehicle-view scenarios by filtering out traffic participants that do not appear in the ego vehicle's multi-view camera images within the expert data. Specifically, we reuse the same assigned ego vehicle and its associated expert trajectory. In this experiment, we filter only vehicle agents while retaining other agent types such as pedestrians and cyclists, and a more comprehensive filtering strategy including all occluded agent types. We then reload the modified, occlusion-prone scenarios into DriveE2E and re-evaluate several baselines.

**Analysis.** As shown in Table 5, AD-MLP and VAD achieve higher driving scores when vehicle agents are removed (AD-MLP: 29.01→29.87; VAD: 62.29→64.17). When all occluded agents are filtered, most baselines—AD-MLP, VAD, and UniAD—improve further (AD-MLP: 29.01→29.57; VAD: 62.29→65.89; UniAD: 77.62→78.49). The exception is TCP Wu et al. (2022), which drops from 48.47 to 46.66. These trends suggest that occlusion-induced incompleteness simplifies closed-loop evaluation by reducing interaction complexity; the effect strengthens as more agents are filtered.

Table 5: More Comparison with Occlusion Filtering. 'Occ.' denotes occlusion.

| Models | DS in Different Benchmarks ↑ | | |
|---|---|---|---|
| | Complete | Occ. Filtering (Vehicle) | Occ. Filtering (All) |
| AD-MLP Zhai et al. (2023) | 29.01 | 29.87 | 29.57 (+0.56) |
| TCP Wu et al. (2022) | 48.47 | 47.53 | 46.66 (-1.79) |
| VAD Jiang et al. (2023) | 62.29 | 64.17 | 65.89 (+3.60) |
| UniAD Hu et al. (2023b) | 77.62 | 76.80 | 78.49 (+0.87) |

## 5 CONCLUSIONS

This work presents DriveE2E, an innovative closed-loop benchmark for advancing end-to-end autonomous driving research by real-to-simulation and offline infrastructure cooperation. By integrating real-world traffic scenarios and static twin road environments into the CARLA simulator, DriveE2E offers a more realistic and reliable evaluation framework that addresses the limitations of both traditional open-loop methods and existing CARLA-based closed-loop evaluations. DriveE2E includes digital twins of 15 diverse urban intersections and 800 traffic scenarios generated from infrastructure sensor data, encompassing various driving behaviors, weather conditions, and times of day. Additionally, we present a robust evaluation benchmark featuring baseline E2EAD methods, enabling comprehensive closed-loop assessments. We believe DriveE2E will greatly contribute to the autonomous driving community and improve the real-world applicability of E2EAD systems.

**Limitations.** Our current setup uses log-replay, so non-ego agents do not react to the ego vehicle, reducing the realism of traffic interactions. We will augment the benchmark with interactive controllers. Visual rendering presently relies on the CARLA engine and thus offers limited fidelity; we plan to leverage generative models to improve realism.

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

Contributors. Carla autonomous driving leaderboard, 2024.

Jiaxun Cui, Hang Qiu, Dian Chen, Peter Stone, and Yuke Zhu. Coopernaut: End-to-end driving with cooperative perception for networked vehicles. In *Proceedings of the IEEE/CVF Conference on Computer Vision and Pattern Recognition*, pages 17252–17262, 2022.

Tianyuan Dai, Josiah Wong, Yunfan Jiang, Chen Wang, Cem Gokmen, Ruohan Zhang, Jiajun Wu, and Li Fei-Fei. Automated creation of digital cousins for robust policy learning. In *Proceedings of the 9th Conference on Robot Learning (CoRL)*, 2025.

Daniel Dauner, Marcel Hallgarten, Tianyu Li, Xinshuo Weng, Zhiyu Huang, Zetong Yang, Hongyang Li, Igor Gilitschenski, Boris Ivanovic, Marco Pavone, et al. Navsim: Data-driven non-reactive autonomous vehicle simulation and benchmarking. *Advances in Neural Information Processing Systems*, 37:28706–28719, 2024.

Alexey Dosovitskiy, German Ros, Felipe Codevilla, Antonio Lopez, and Vladlen Koltun. Carla: An open urban driving simulator. In *Conference on robot learning*, pages 1–16, 2017.

Daocheng Fu, Xin Li, Licheng Wen, Min Dou, Pinlong Cai, Botian Shi, and Yu Qiao. Drive like a human: Rethinking autonomous driving with large language models. In *Proceedings of the IEEE/CVF Winter Conference on Applications of Computer Vision*, pages 910–919, 2024.

Mike Goslin and Mark R Mine. The panda3d graphics engine. *Computer*, 37(10):112–114, 2004.

Ruiyang Hao, Siqi Fan, Yingru Dai, Zhenlin Zhang, Chenxi Li, Yuntian Wang, Haibao Yu, Wenxian Yang, Jirui Yuan, and Zaiqing Nie. Rcooper: A real-world large-scale dataset for roadside cooperative perception. In *Proceedings of the IEEE/CVF Conference on Computer Vision and Pattern Recognition*, pages 22347–22357, 2024.

Ruiyang Hao, Bowen Jing, Haibao Yu, and Zaiqing Nie. Styledrive: Towards driving-style aware benchmarking of end-to-end autonomous driving. *arXiv preprint arXiv:2506.23982*, 2025a.

Ruiyang Hao, Haibao Yu, Jiaru Zhong, Chuanye Wang, Jiahao Wang, Yiming Kan, Wenxian Yang, Siqi Fan, Huilin Yin, Jianing Qiu, et al. Research challenges and progress in the end-to-end v2x cooperative autonomous driving competition. *arXiv preprint arXiv:2507.21610*, 2025b.

Anthony Hu, Lloyd Russell, Hudson Yeo, Zak Murez, George Fedoseev, Alex Kendall, Jamie Shotton, and Gianluca Corrado. Gaia-1: A generative world model for autonomous driving. *arXiv preprint arXiv:2309.17080*, 2023a.

Yihan Hu, Jiazhi Yang, Li Chen, Keyu Li, Chonghao Sima, Xizhou Zhu, Siqi Chai, Senyao Du, Tianwei Lin, Wenhai Wang, et al. Planning-oriented autonomous driving. In *Proceedings of the IEEE/CVF Conference on Computer Vision and Pattern Recognition*, pages 17853–17862, 2023b.

Bernhard Jaeger, Kashyap Chitta, and Andreas Geiger. Hidden biases of end-to-end driving models. In *Proceedings of the IEEE/CVF International Conference on Computer Vision*, pages 8240–8249, 2023.

Xiaosong Jia, Penghao Wu, Li Chen, Jiangwei Xie, Conghui He, Junchi Yan, and Hongyang Li. Think twice before driving: Towards scalable decoders for end-to-end autonomous driving. In *Proceedings of the IEEE/CVF Conference on Computer Vision and Pattern Recognition*, pages 21983–21994, 2023.

Xiaosong Jia, Zhenjie Yang, Qifeng Li, Zhiyuan Zhang, and Junchi Yan. Bench2drive: Towards multi-ability benchmarking of closed-loop end-to-end autonomous driving. *arXiv preprint arXiv:2406.03877*, 2024.

Bo Jiang, Shaoyu Chen, Qing Xu, Bencheng Liao, Jiajie Chen, Helong Zhou, Qian Zhang, Wenyu Liu, Chang Huang, and Xinggang Wang. Vad: Vectorized scene representation for efficient autonomous driving. In *Proceedings of the IEEE/CVF International Conference on Computer Vision*, pages 8340–8350, 2023.

Alex Kendall, Jeffrey Hawke, David Janz, Przemyslaw Mazur, Daniele Reda, John-Mark Allen, Vinh-Dieu Lam, Alex Bewley, and Amar Shah. Learning to drive in a day. In *2019 International Conference on Robotics and Automation (ICRA)*, pages 8248–8254. IEEE, 2019.

Quanyi Li, Zhenghao Peng, Lan Feng, Qihang Zhang, Zhenghai Xue, and Bolei Zhou. Metadrive: Composing diverse driving scenarios for generalizable reinforcement learning. *IEEE transactions on pattern analysis and machine intelligence*, 45(3):3461–3475, 2022a.

Quanyi Li, Zhenghao Mark Peng, Lan Feng, Zhizheng Liu, Chenda Duan, Wenjie Mo, and Bolei Zhou. Scenarionet: Open-source platform for large-scale traffic scenario simulation and modeling. *Advances in neural information processing systems*, 36:3894–3920, 2023.

Yingyan Li, Lue Fan, Jiawei He, Yuqi Wang, Yuntao Chen, Zhaoxiang Zhang, and Tieniu Tan. Enhancing end-to-end autonomous driving with latent world model. *arXiv preprint arXiv:2406.08481*, 2024a.

Zhiqi Li, Wenhai Wang, Hongyang Li, Enze Xie, Chonghao Sima, Tong Lu, Yu Qiao, and Jifeng Dai. Bevformer: Learning bird's-eye-view representation from multi-camera images via spatiotemporal transformers. In *European conference on computer vision*, pages 1–18. Springer, 2022b.

Zhiqi Li, Zhiding Yu, Shiyi Lan, Jiahan Li, Jan Kautz, Tong Lu, and Jose M Alvarez. Is ego status all you need for open-loop end-to-end autonomous driving? In *Proceedings of the IEEE/CVF Conference on Computer Vision and Pattern Recognition*, pages 14864–14873, 2024b.

Xiaodan Liang, Tairui Wang, Luona Yang, and Eric Xing. Cirl: Controllable imitative reinforcement learning for vision-based self-driving. In *Proceedings of the European conference on computer vision (ECCV)*, pages 584–599, 2018.

Genjia Liu, Yue Hu, Chenxin Xu, Weibo Mao, Junhao Ge, Zhengxiang Huang, Yifan Lu, Yinda Xu, Junkai Xia, Yafei Wang, et al. Towards collaborative autonomous driving: Simulation platform and end-to-end system. *arXiv preprint arXiv:2404.09496*, 2024.

MathWorks. Roadrunner, 2023.

OSM contributors. Openstreetmap: The free wiki world map, 2023.

Chenbin Pan, Burhaneddin Yaman, Tommaso Nesti, Abhirup Mallik, Alessandro G Allievi, Senem Velipasalar, and Liu Ren. Vlp: Vision language planning for autonomous driving. In *Proceedings of the IEEE/CVF Conference on Computer Vision and Pattern Recognition*, pages 14760–14769, 2024.

Aditya Prakash, Kashyap Chitta, and Andreas Geiger. Multi-modal fusion transformer for end-to-end autonomous driving. In *Proceedings of the IEEE/CVF conference on computer vision and pattern recognition*, pages 7077–7087, 2021.

Danila Rukhovich, Anna Vorontsova, and Anton Konushin. Imvoxelnet: Image to voxels projection for monocular and multi-view general-purpose 3d object detection. In *Proceedings of the IEEE/CVF Winter Conference on Applications of Computer Vision*, pages 2397–2406, 2022.

Hao Shao, Letian Wang, Ruobing Chen, Hongsheng Li, and Yu Liu. Safety-enhanced autonomous driving using interpretable sensor fusion transformer. In *Conference on Robot Learning*, pages 726–737, 2023a.

Hao Shao, Letian Wang, Ruobing Chen, Steven L Waslander, Hongsheng Li, and Yu Liu. Reasonnet: End-to-end driving with temporal and global reasoning. In *Proceedings of the IEEE/CVF conference on computer vision and pattern recognition*, pages 13723–13733, 2023b.

Hao Shao, Yuxuan Hu, Letian Wang, Guanglu Song, Steven L Waslander, Yu Liu, and Hongsheng Li. Lmdrive: Closed-loop end-to-end driving with large language models. In *Proceedings of the IEEE/CVF Conference on Computer Vision and Pattern Recognition*, pages 15120–15130, 2024.

Chonghao Sima, Katrin Renz, Kashyap Chitta, Li Chen, Hanxue Zhang, Chengen Xie, Ping Luo, Andreas Geiger, and Hongyang Li. Drivelm: Driving with graph visual question answering. In *European Conference on Computer Vision*, 2024.

Ziying Song, Caiyan Jia, Lin Liu, Hongyu Pan, Yongchang Zhang, Junming Wang, Xingyu Zhang, Shaoqing Xu, Lei Yang, and Yadan Luo. Don't shake the wheel: Momentum-aware planning in end-to-end autonomous driving. In *Proceedings of the Computer Vision and Pattern Recognition Conference*, pages 22432–22441, 2025.

Adam Tonderski, Carl Lindström, Georg Hess, William Ljungbergh, Lennart Svensson, and Christoffer Petersson. Neurad: Neural rendering for autonomous driving. In *Proceedings of the IEEE/CVF Conference on Computer Vision and Pattern Recognition*, pages 14895–14904, 2024.

Martin Treiber, Ansgar Hennecke, and Dirk Helbing. Congested traffic states in empirical observations and microscopic simulations. *Physical review E*, 62(2):1805, 2000.

Xiaofeng Wang, Zheng Zhu, Guan Huang, Xinze Chen, Jiagang Zhu, and Jiwen Lu. Drivedreamer: Towards real-world-driven world models for autonomous driving. *arXiv preprint arXiv:2309.09777*, 2023.

Xinshuo Weng, Jianren Wang, David Held, and Kris Kitani. Ab3dmot: A baseline for 3d multi-object tracking and new evaluation metrics. *arXiv preprint arXiv:2008.08063*, 2020.

Penghao Wu, Xiaosong Jia, Li Chen, Junchi Yan, Hongyang Li, and Yu Qiao. Trajectory-guided control prediction for end-to-end autonomous driving: A simple yet strong baseline. In *Advances in Neural Information Processing Systems*, pages 6119–6132, 2022.

Chejian Xu, Wenhao Ding, Weijie Lyu, Zuxin Liu, Shuai Wang, Yihan He, Hanjiang Hu, Ding Zhao, and Bo Li. Safebench: A benchmarking platform for safety evaluation of autonomous vehicles. *Advances in Neural Information Processing Systems*, 35:25667–25682, 2022.

Zhenhua Xu, Yujia Zhang, Enze Xie, Zhen Zhao, Yong Guo, Kwan-Yee K Wong, Zhenguo Li, and Hengshuang Zhao. Drivegpt4: Interpretable end-to-end autonomous driving via large language model. *IEEE Robotics and Automation Letters*, pages 1–8, 2024.

Xuemeng Yang, Licheng Wen, Yukai Ma, Jianbiao Mei, Xin Li, Tiantian Wei, Wenjie Lei, Daocheng Fu, Pinlong Cai, Min Dou, et al. Drivearena: A closed-loop generative simulation platform for autonomous driving. *arXiv preprint arXiv:2408.00415*, 2024.

Haibao Yu, Yizhen Luo, Mao Shu, Yiyi Huo, Zebang Yang, Yifeng Shi, Zhenglong Guo, Hanyu Li, Xing Hu, Jirui Yuan, et al. Dair-v2x: A large-scale dataset for vehicle-infrastructure cooperative 3d object detection. In *Proceedings of the IEEE/CVF Conference on Computer Vision and Pattern Recognition*, pages 21361–21370, 2022.

Haibao Yu, Wenxian Yang, Jiaru Zhong, Zhenwei Yang, Siqi Fan, Ping Luo, and Zaiqing Nie. End-to-end autonomous driving through v2x cooperation. *arXiv preprint arXiv:2404.00717*, 2024.

Jiang-Tian Zhai, Ze Feng, Jinhao Du, Yongqiang Mao, Jiang-Jiang Liu, Zichang Tan, Yifu Zhang, Xiaoqing Ye, and Jingdong Wang. Rethinking the open-loop evaluation of end-to-end autonomous driving in nuscenes. *arXiv preprint arXiv:2305.10430*, 2023.

Wenzhao Zheng, Ruiqi Song, Xianda Guo, Chenming Zhang, and Long Chen. Genad: Generative end-to-end autonomous driving. *arXiv preprint arXiv:2402.11502*, 2024.

## VI    COMPARISON WITH OPEN-LOOP AND CLOSED-LOOP EVALUATION

To clarify the position of DriveE2E, we compare it against the commonly used open-loop and closed-loop evaluation protocols along two key dimensions: how other agents are controlled and how ego observations are obtained. Furthermore, we summarize their respective advantages and limitations. As shown in Table 6, DriveE2E adopts a hybrid setting where non-ego agents follow log-replayed real trajectories while the ego vehicle receives simulation-generated observations, thereby combining the realism and reproducibility of open-loop with the interaction capability of closed-loop.

| Evaluation Type | Other Agents | Ego Observation | Advantages | Limitations |
|---|---|---|---|---|
| Open-loop | Log-replay | Log-replay | Efficient; uses real trajectories; simple to implement | No interaction; limited for long-horizon planning |
| Closed-loop (typical) | Algorithm control | Simulation-generated | Captures interaction; realistic online decision-making | Scenario design often manual; reactive models may be unstable or unrealistic |
| DriveE2E (ours) | Log-replay | Simulation-generated | High-fidelity realism from real-world trajectories; reproducible and efficient | Other agents do not react to ego vehicle (log-replay only) |

Table 6: Comparison of different evaluation protocols for E2EAD.

## VII    EVALUATION METRICS

We adopt two metrics to evaluate the performance of the E2EAD system:

- **Success Rate (SR).** This metric measures the percentage of successfully completed routes within a certain time without collisions or traffic violations (e.g., leaving the drivable area).

- **Driving Score (DS).** This metric measures the driving performance while taking the route completion $RC_i$ and infraction penalty of $i$-route into account as Eq. 1.

$$DS = \frac{1}{n_{total}} \sum_{i=1}^{n_{total}} RC_i \prod_{j=1}^{inf_i} (p_i^j),  \tag{1}$$

where $n_{total}$ denotes the total number of routes, $inf_i$ means a set of infraction that the ego vehicle triggered in $i$th-route, and $p_i^j$ denotes the infraction penalty coefficient. For more details about infraction types and coefficients, refer to CARLA LB V2 Contributors (2024).

## VIII    BASELINE MODELS

We train and evaluate the following models in our DriveE2E benchmark:

- UniAD Hu et al. (2023b) employs queries to integrate key tasks such as perception, mapping, prediction, and planning. The standard training process for UniAD typically involves three stages. To accelerate training and reduce GPU resource consumption, we bypassed the initial stages by directly training the stage-2 model using the BEVFormer Li et al. (2022b) model provided by Bench2Drive Jia et al. (2024) as a pre-trained model. We trained UniAD for one epoch. It is important to note that these settings may lead to a reduction in UniAD's accuracy.

- VAD Jiang et al. (2023) employs Transformer queries while enhancing efficiency through a vectorized scene representation. We trained the VAD model for two epochs, using a pre-trained model provided by Bench2Drive Jia et al. (2024) as a pretrain.

- AD-MLP Zhai et al. (2023) adopts a simple strategy by using the ego-vehicle past states into an MLP to generate future trajectory predictions.

- TCP Wu et al. (2022) predicts both trajectories and control signals. It only uses front-facing cameras and the ego state as inputs. Note that we did not train an expert model and did not use expert feature distillation during TCP training.

- MomAD Song et al. (2025) introduces trajectory momentum and perception momentum to stabilize and refine trajectory predictions, finally enhance the planning performance.

## IX ADDITIONAL RESULTS: OPEN-LOOP VS. CLOSED-LOOP

In this section, we evaluate different UniAD models using both open-loop and closed-loop approaches. We provide extra collision rate evaluation results for open-loop evaluation. Specifically, we save intermediate checkpoint models during the training of UniAD and assess these checkpoint models through both open-loop and closed-loop evaluations.

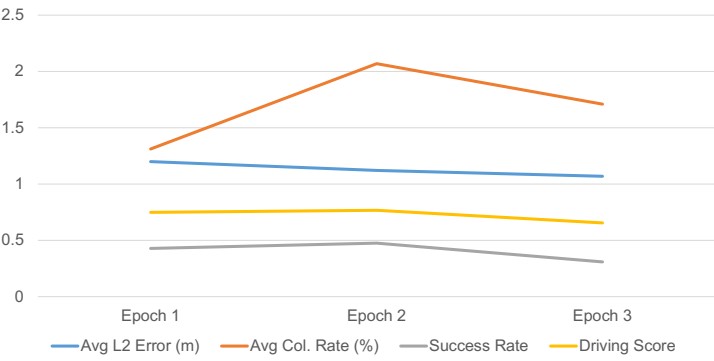

Figure 5: **UniAD's Evaluation Results.**

From Figure 5, we observe that while L2 error trends in open-loop evaluation differ from success rate and driving score in closed-loop evaluation, the fluctuations are minimal. In contrast, the collision rate aligns with success rate and driving score but fluctuates significantly more, highlighting the importance of closed-loop evaluations.

## X STATIC INTERSECTION CONSTRUCTION DETAILS

### X.1 CONSTRUCTION PROCESS

We provide more details to illustrate how to construct the static intersection assets in Figure 6. Here RoadRunner MathWorks (2023) is a 3D environment editing tool used for designing and editing road and traffic scenes for simulation and testing of autonomous driving systems. OpenStreetMap OSM contributors (2023) is a global, user-contributed, open-source map database. Blender Blender Studio (2023) is an open-source 3D creation suite for modeling, animation, and rendering.

### X.2 INTERSECTION ASSETS VISUALIZATION

DriveE2E presents 15 digital twins of urban intersections, each carefully designed to incorporate detailed roadside and road features, including traffic light poles, signage, lanes, crosswalks, stop lines, and surrounding buildings. These constructed twin intersections are presented in Figure 7.

## XI SIMULATION-TO-REAL (SIM2REAL) DISCUSSION

We examine how DriveE2E addresses the Sim2Real gap from three perspectives: fidelity of digital-twin scenarios, perception performance when evaluated on real-world datasets, and end-to-end model performance in real-world driving.

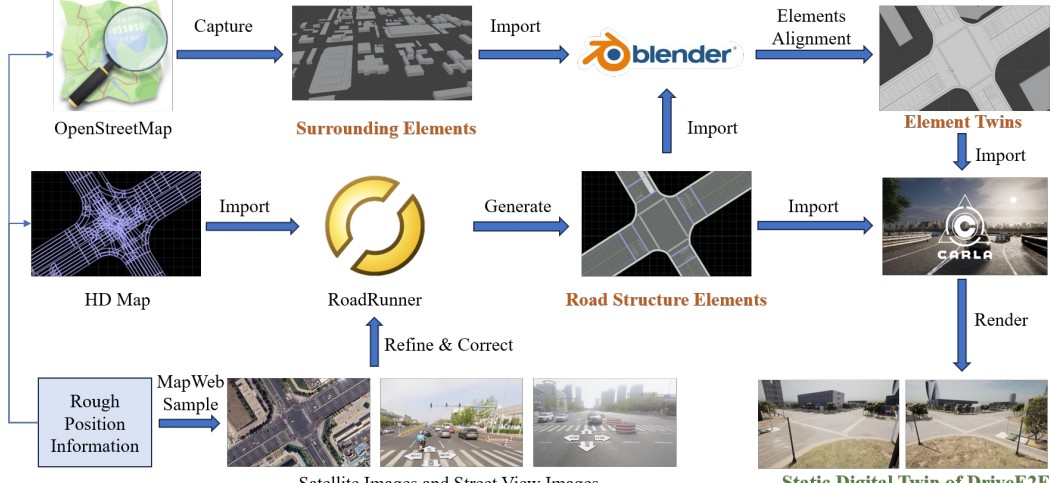

Figure 6: **Static Intersections Assets Construction Process**: We first obtain HD maps for each intersection and refine the road structures in RoadRunner. Surrounding elements are collected from OpenStreetMap. Finally, we integrate all components—including traffic light poles and signals—using Blender to create static intersection assets compatible with CARLA Dosovitskiy et al. (2017). These assets can be rendered directly in the CARLA simulator.

## XI.1 FIDELITY COMPARISON

Our digital-twin driving scenarios achieve high fidelity to real-world intersections across several dimensions, including:

- Static Road Environment: We replicate road layouts from HD maps with fine-grained details such as markings, vegetation, and surrounding buildings, ensuring close alignment with the real world. To quantify this, we compute element-wise similarity metrics between our digital assets and satellite imagery (Table 7a).

- Traffic Agent Appearance: Vehicles, cyclists, and pedestrians are matched with the most visually similar CARLA assets, achieving high similarity in both size and appearance (Table 7b).

- Traffic Behaviors: Dynamic agents follow strictly recorded real-world trajectories, preserving authentic interactions.

- Traffic Lights: Signal states and timings are reproduced from real-world logs with 100% consistency.

- Environmental Conditions: Lighting and weather effects are simulated with up to 97% similarity, with future refinements planned using sensor data.

| Type | Similarity (%) |
|------|----------------|
| Lane | 91 |
| Crosswalk | 86 |
| Road Mark | 90 |
| Surrounding Building | 50 |
| Plant | 80 |

(a) Road and surrounding elements

| Type | Similarity (%) |
|------|----------------|
| Vehicle | 92 |
| Cyclist | 88 |
| Pedestrian | 89 |

(b) Agent appearance

Table 7: Similarity statistics for different categories in Sim2Real comparison.

## XI.2 PERCEPTION PERFORMANCE DEGRADATION

We evaluate perception consistency using BEVFormer (UniAD stage-1). Comparisons are made between our DriveE2E expert dataset and the real-world nuScenes dataset. For fairness, DriveE2E's

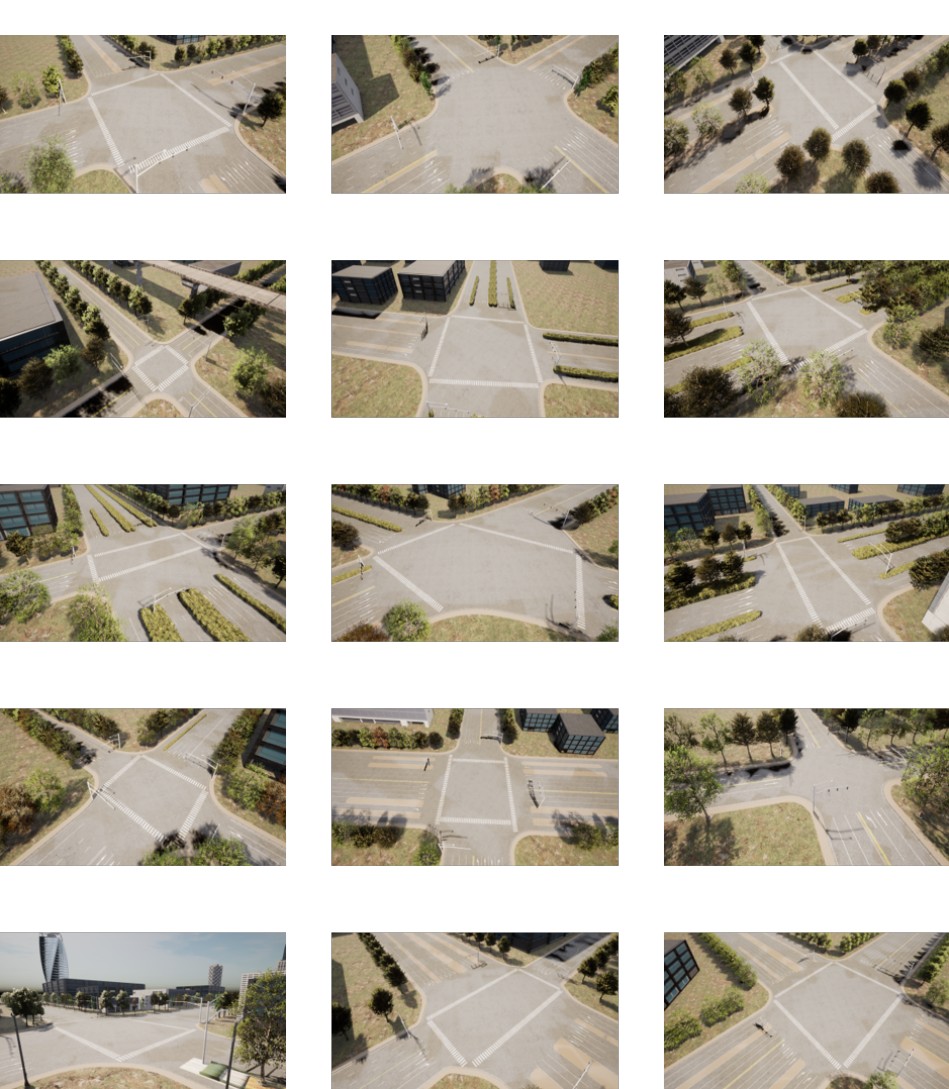

Figure 7: **Visualization of 15 Twin Intersection Assets.**. These twins encompass intricate roadside elements, including traffic light poles and nearby buildings, along with diverse road features such as signage, lanes, crosswalks, and stop lines.

sensors are configured to match nuScenes, as (1) our own collected real-world dataset uses different sensor setups, and (2) large-scale annotation would be prohibitively costly. We report results in Table 8, where **BEVFormer-DriveE2E** denotes training on the DriveE2E expert dataset and **BEVFormer-nuScenes** denotes training on nuScenes. The former is tested on both datasets, while the latter is evaluated only on nuScenes.

| Model | Dataset | 3D Object Detection (Vehicle) |
|---|---|---|
| BEVFormer-DriveE2E | DriveE2E | 0.632 |
| BEVFormer-nuScenes | nuScenes | 0.591 |
| BEVFormer-DriveE2E | nuScenes | 0.129 |

Table 8: **Sim2Real Comparison for Perception Performance.** Here we provide the 3D object detection results (vehicle category).

From Table 8, we observe that BEVFormer trained on DriveE2E exhibits a significant drop when evaluated on nuScenes. This degradation stems from both the *sensor data domain gap* and the *environmental domain gap*, with the former being dominant. Differences extend beyond FOV or lens distortion to include ISP type, compression methods, resolution, and frame rate. Notably, such gaps appear not only in *simulation-to-real* but also across *real-to-real* datasets collected with different vehicles and sensors. For instance, two autonomous vehicles with distinct sensor suites driving through the same intersection will often exhibit degraded cross-domain generalization. These gaps affect both perception and downstream planning, complicating deployment of models across sensor domains.

To address this, DriveE2E provides an expert Dataset collected under strictly controlled sensor configurations. Results show that when training and testing use the *same type of sensor data*, relative model performance is preserved, confirming that DriveE2E is a fair benchmark for perception.

### XI.3    END-TO-END MODEL PERFORMANCE IN REAL-WORLD TESTING

We have not yet conducted controlled real-world closed-loop end-to-end evaluation due to two primary challenges.

- Reproducibility difficulty. Unlike tabletop Sim2Real experiments Dai et al. (2025), real-world driving intersections are dynamic and cannot guarantee identical participants, trajectories, or interactions across trials. This makes it infeasible to conduct fair, repeatable comparisons of end-to-end models. Artificially controlling traffic would also pose *safety concerns* for other road users.
- Deployment difficulty. Deploying a trained PyTorch model from DriveE2E into a real vehicle is non-trivial, requiring: (1) engineering integration into onboard computing with real-time safety guarantees; (2) resolving sensor mismatches between DriveE2E and real vehicles; and (3) bridging the control gap between CARLA's internal API and real-world actuation interfaces.

### XI.4    SUMMARY

DriveE2E provides high-fidelity digital twins that enable reproducible, fair benchmarking of autonomous driving models. While real-world closed-loop tests remain impractical, DriveE2E's real2sim pipeline ensures that research remains well-aligned with real-world driving. Models performing strongly in this environment can later be adapted and validated by industry partners using their own deployment pipelines.

## XII    DRIVING SCENARIOS VISUALIZATION

**Driving Behavior Illustration.**    DriveE2E identifies and categorizes eight distinct driving scenarios from 800 real-world traffic clips, capturing typical driving behaviors at intersections. These scenarios include Interaction with Pedestrians and Cyclists (IPC), Competing with Other Vehicles (COV), Passing Through During Yellow Lights (YLW), Making a U-turn (UT), Stopping at Red Lights (STP), Going Straight Through Intersections (STR), Making a Left Turn (LFT), and Making a Right Turn (RT).

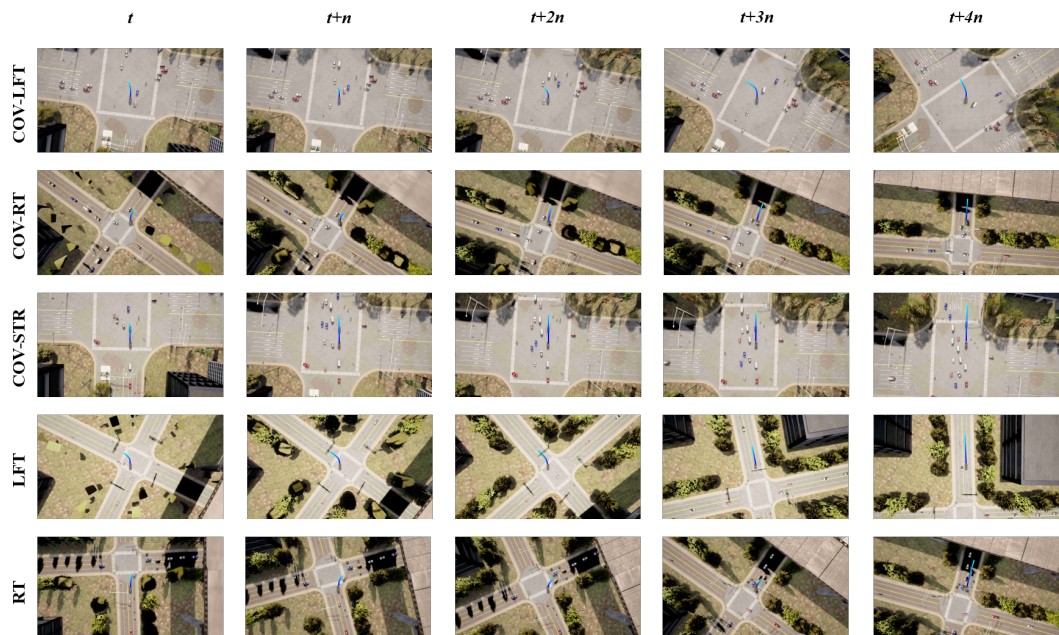

Figure 8: **Five Sub-scenario Driving Behavior Visualization:** This visualization encompasses five driving scenarios: competing with other vehicles while turning left (COV-LET), turning right (COV-RT), and going straight (COV-STR), as well as normal left turns (LFT) and right turns (RT). Frames are sampled at intervals $t$, $t + n$, $t + 2n$, $t + 3n$, and $t + 4n$ from the driving sequences to depict the vehicle's behavior over time. Each image is presented from a **top-down view**, with the ego vehicle (depicted in gray) centrally positioned. The vehicle's motion direction is represented by a purple trajectory line.

- **IPC**: *Interaction with Pedestrians and Cyclists* involves safely navigating around or yielding to pedestrians and cyclists.

- **COV**: *Competing with Other Vehicles* refers to scenarios where the vehicle asserts its position in traffic, such as during merges or unprotected left turns.

- **YLW**: *Passing through during Yellow Lights* describes the decision-making process of whether to stop or start when the light turns yellow, balancing safety and timing.

- **UT**: *Making a U-turn* involves turning the vehicle to reverse its direction, either partially or fully, at an intersection or designated point.

- **STP**: *Stopping at Red Lights* involves halting the vehicle to comply with traffic signals.

- **STR, LFT, RT**: *Going Straight through Intersection, Making a Left Turn, and Making a Right Turn* are the most common driving behaviors at intersections, not specifically categorized under the other types.

These eight scenarios are **further refined into 14 specific sub-scenarios** based on turning conditions and anomalies. We illustrate these sub-scenarios in Figure 8, Figure 9 and Figure 10.

**Twin Weather and Lighting Conditions.** We meticulously document the weather and lighting conditions for each driving scenario, enabling DriveE2E to accurately reconstruct these elements as they originally appeared. Specifically, weather data and timestamps were recorded during the capture of original infrastructure sensor data. This approach allows us to replicate the precise weather states and lighting angles within the CARLA simulator using its built-in weather system. To illustrate the twin effects, we present a reconstructed scene under diverse weather and lighting conditions in Figure 11.

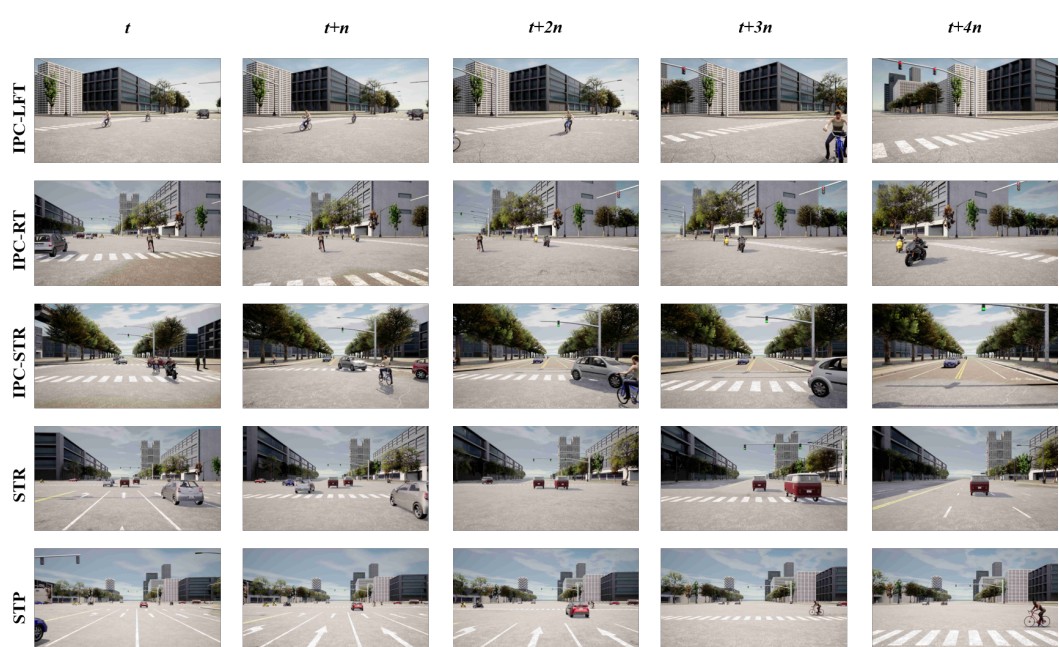

Figure 9: **Five Sub-scenario Driving Behavior Visualization:** This visualization encompasses five driving scenarios: Interaction with pedestrians and cyclists while turning left (IPC-LET), turning right (IPC-RT), and going straight (IPC-STR), along with normal straight driving (STR) and stopping at red lights (STP). Each image is presented from a **front view**.

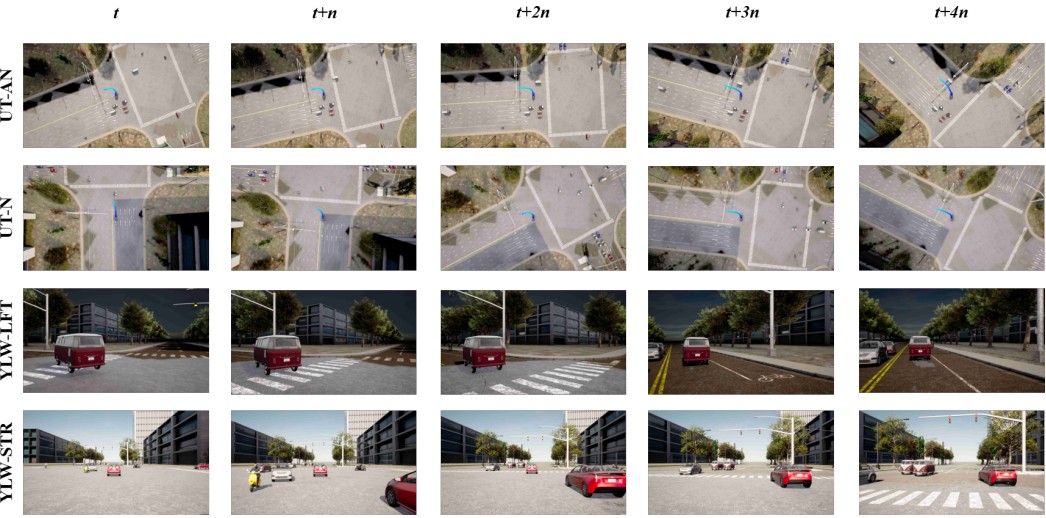

Figure 10: **Four Sub-scenario Driving Behavior Visualization:** This visualization encompasses four driving scenarios: U-turns in abnormal (UT-AN) and normal conditions (UT-N), and passing through yellow lights while turning left (YLW-LFT) or going straight (YLW-STR). Each image is presented from a **top-down view** or **front-head view**.

## XIII   TRAFFIC PARTICIPANTS MISSING VISUALIZATION

We also present a visualization comparing traffic scenarios before and after occlusion-based agent filtering in Figure 12 and Figure 13.

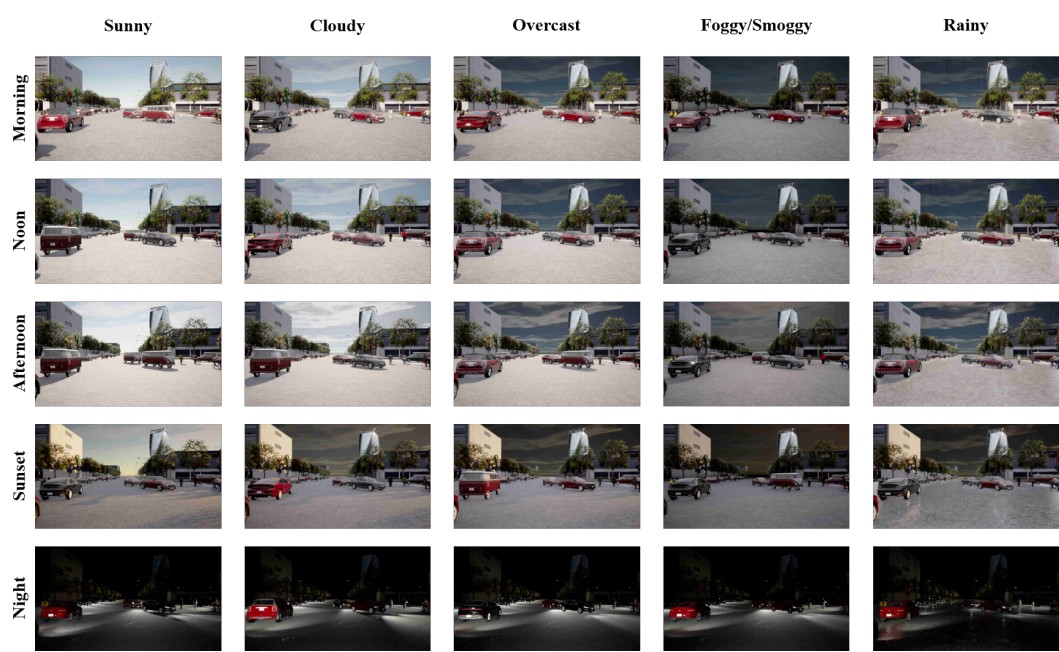

Figure 11: Twin Weather and Light Conditions Visualization.

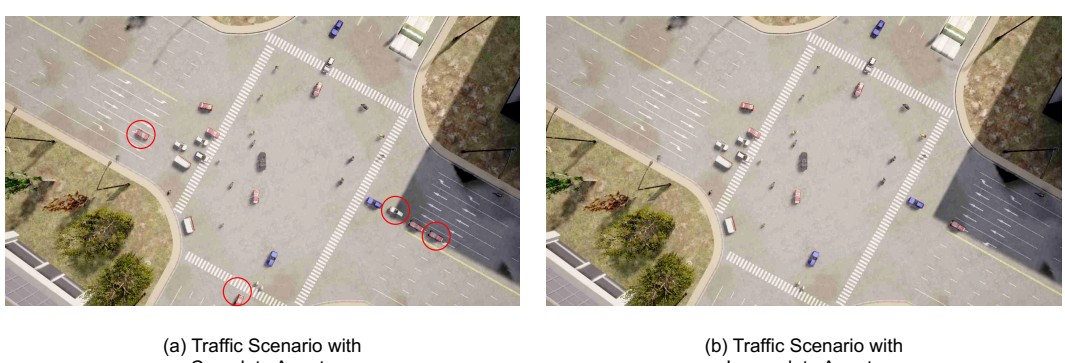

(a) Traffic Scenario with
Complete Agents

(b) Traffic Scenario with
Incomplete Agents

Figure 12: **Traffic Scenario Comparison with Occlusion Filtering:** (a) Traffic scenario extracted from infrastructure sensor data, capturing all traffic agents. (b) Traffic scenario constructed by filtering out agents occluded from the ego-vehicle's sensor view. In this comparison, only vehicle agents are filtered, while all other agent types are retained.

## XIV PLANNING RESULTS VISUALIZATION

This section presents the visualization of planning results, showcasing both successful and failed cases of the VAD model across three scenarios: competing with other vehicles (COV), normal left turns (LFT), and going straight (STR). The corresponding visualizations are shown in Figure 14, Figure 15, and Figure 16, respectively.

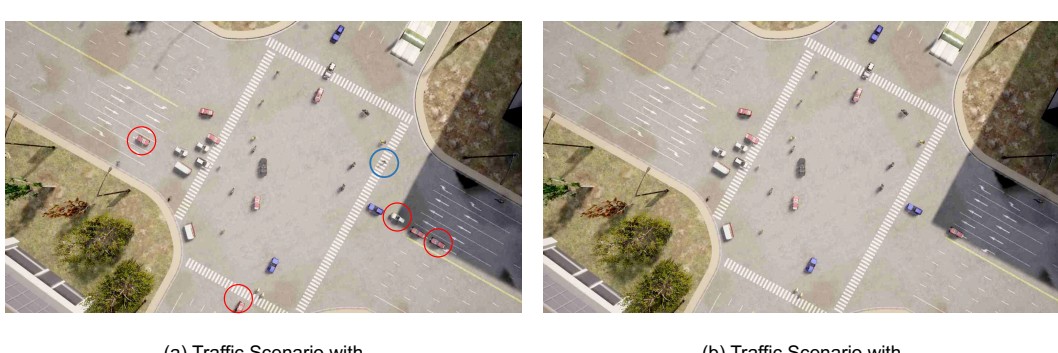

(a) Traffic Scenario with
Complete Agents

(b) Traffic Scenario with
Incomplete Agents (Filter All)

Figure 13: **Traffic Scenario Comparison with All Occluded-Agents Filtering:** (a) Traffic scenario extracted from infrastructure sensor data, capturing all traffic agents. (b) Traffic scenario constructed by filtering out agents occluded from the ego-vehicle's sensor view. In this comparison, all agents are filtered.

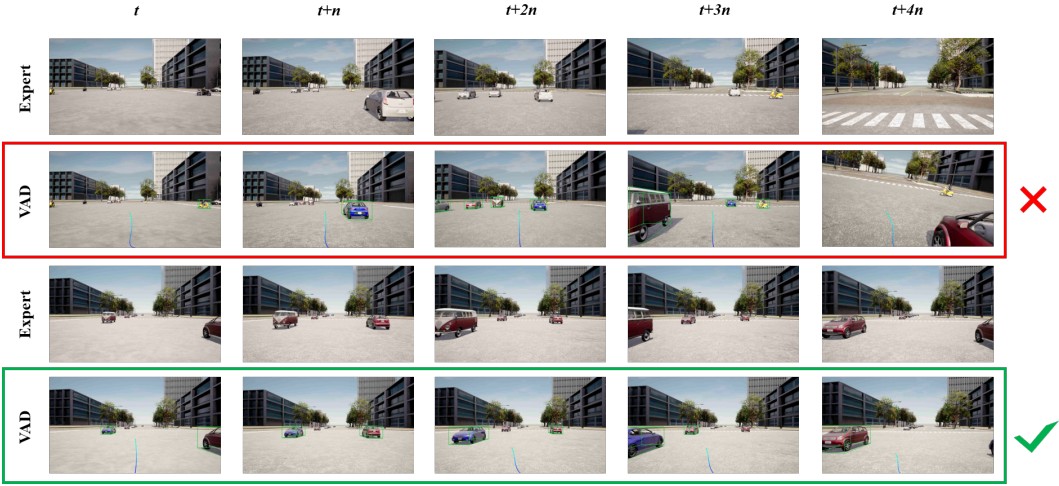

Figure 14: **Successful and Failed Cases Behaved in the COV Scenario:** In the failed case, the ego vehicle controlled by the trained VAD model exhibited excessive caution while competing for the lane with another vehicle, failing to account for a car approaching from the right rear. This oversight led to a collision due to the ego vehicle's slow speed. Conversely, the successful case demonstrated effective lane competition at a reasonable speed, avoiding any collisions.

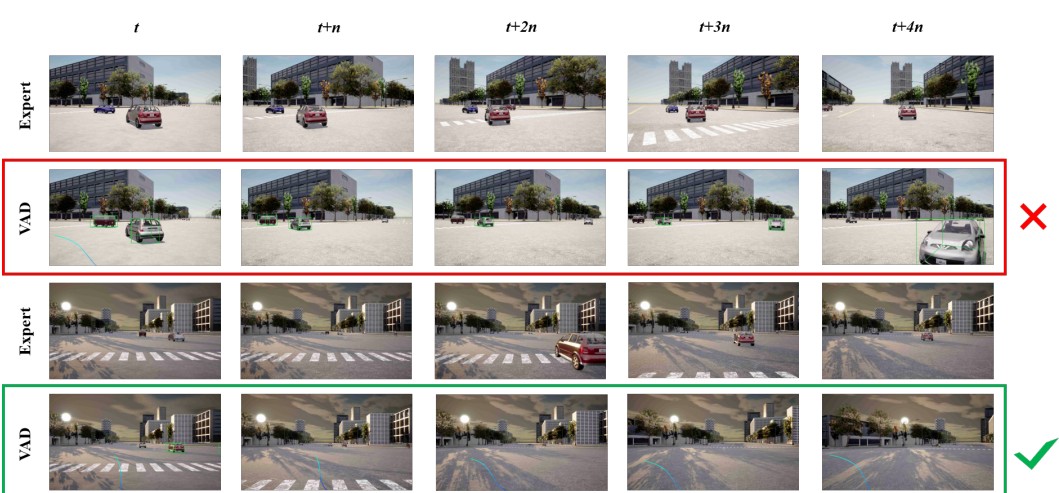

Figure 15: **Successful and Failed Cases Behaved in the LFT Scenarios:** In the failed case, the VAD-controlled ego vehicle was overly cautious during a left turn and failed to anticipate oncoming traffic, leading to a collision. In contrast, the successful case demonstrated a smooth and well-timed turn, avoiding any interference from oncoming vehicles.

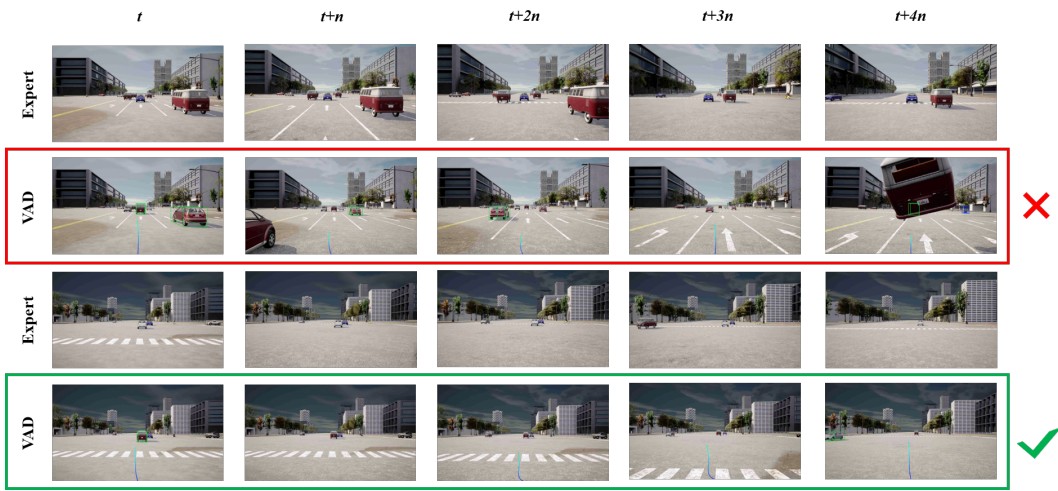

Figure 16: **Successful and Failed Cases Behaved in the STR Scenarios:** In the failed case, the ego vehicle controlled by the trained VAD model accelerated too slowly while traveling straight, leading to a collision with a trailing vehicle. In contrast, the successful case demonstrated the VAD-controlled ego vehicle navigating the intersection smoothly at an appropriate speed, avoiding any collisions.

