# OpenReview forum: "DriveE2E: Closed-Loop Benchmark for End-to-End Autonomous Driving through Real-to-Simulation"
_ICLR.cc/2026/Conference — Submitted to ICLR 2026_

### Official Review · Reviewer_2DSC · 2025-10-17

**Soundness:** 2
**Presentation:** 3
**Contribution:** 2
**Rating:** 4
**Confidence:** 5

**Summary:**

This paper proposes a novel closed-loop evaluation benchmark of end-to-end autonomous driving agents by constructing digital twins of real-world intersections and replaying logs of the traffic participants. It uses infrastructure sensors for reconstruction and the CARLA simulator for simulation. The benchmark is comprehensive and consists of various driving behaviors and traffic participants.

**Strengths:**

1. Closed-loop simulation is important for benchmarking end-to-end driving models and constructing digital twins of real-world driving scenes can help with closed-loop testing and bridge the sim2real gap.
2. Using infrastructure-mounted sensors can help improve the accuracy and complexity of the reconstructed traffic scenario, improving the realism of the digital twins.
3. The proposed DriveE2E benchmark is comprehensive in the number of scenarios, agent behaviors, and traffic participant categories.

**Weaknesses:**

1. Although acknowledged in the limitations section, the log-replay simulation is a major weakness of this work for more realistic closed-loop evaluation, as it should be straightforward to incorporate behavior models for other agents. This could have been an advantage of using a driving simulator like CARLA, which offers easier agent management.
2. The reconstructed digital twin of real-world interactions lacks diversity and realism. The layout of the intersection is all four-way intersections, and the assets, such as bushes, trees, and road textures, all appear the same.
3. The collected real-world data all come from a small region with only 15 intersections, which lacks diversity. Therefore, the results from such a benchmark may not be general and comprehensive for the driving agent's performance.
4. With the lack of photorealism of the image rendering of CARLA, the proposed DriveE2E benchmark is more suitable for modular testing of AD systems like downstream planning and decision making rather than evaluating end-to-end driving agents. End-to-end driving policies trained from such data may also exhibit a large sim2real gap.

**Questions:**

1. Can the authors provide some video results of the driving scenario visualizations?
2. Can the authors provide more statistics on the assets used for creating the digital twins, or do they simply reuse the CARLA assets? Like how many different types of vehicles, pedestrians, etc? Are they fixed or randomly initialized in each scenario?
3. Can the authors elaborate more on the real-world data collection? For example, what types and how many infrastructure sensors are used? What is their perception range?
4. In Tab. 7, can the authors explain more about the element-wise similarity metrics used to evaluate the reconstruction fidelity?

---

> ### Author Response · Authors · 2025-11-27
> **Rebuttal by Authors**
>
> We appreciate the reviewer's insightful comments and valuable feedback.
> # Response to Weakness 1:  Log-replay
> While log-replay limits interactive agent behaviors, it provides a uniquely challenging setting: surrounding agents do not yield or adapt, forcing the ego to plan safely without relying on expert responses. As discussed in **Response to Reviewer XQzd (Weakness 2)**, log-replay was a pragmatic choice for dense urban intersections with many participants, where even simple reactive models can behave unrealistically and advanced models incur high computational costs.
>
> Log-replay also aligns with our digital-twin philosophy: DriveE2E reconstructs 15 real intersections with HD maps and precise trajectories from infrastructure sensors, faithfully capturing road geometry, traffic lights, features, and actual traffic flows. We agree that integrating interactive controllers is valuable and plan to extend the benchmark with multi-agent interactions and CARLA support in future releases.
>
> # Response to Weakness 2-3: Illustration of scenario diversity and realism
> Our benchmark is built on real-world data from infrastructure-side sensors, offering an unobstructed, comprehensive view of intersections and complete multi-agent trajectories, which are essential for evaluating complex urban driving. Current traffic annotations come from surveillance videos, so the release mainly covers intersection scenarios.
>
> During reconstruction, we focused on **road-critical elements**—lane markings, crosswalks, curbs, and traffic signals—while **deprioritizing decorative or visually diverse assets** with limited impact on driving functionality.
>
> For dense-traffic evaluation, **realistic dynamic agent behaviors** provide more meaningful challenges than simply increasing scene coverage. Our benchmark **delivers rich traffic dynamics and authentic trajectories**, supporting robust assessment of autonomous driving systems. We recognize the value of scene diversity and are extending the benchmark: maps for Y-junctions and roundabouts, where sensors are deployed, have been constructed and will appear in future releases to improve variety and generalization.
>
> # Response to Weakness 4: Intrisic shortcoming of image rendering in CARLA
> As noted in our Response to **Reviewer o5n8 (Weakness 2)**, BEVFormer trained and tested on DriveE2E achieves comparable 3D vehicle detection to nuScenes (0.632 vs. 0.591), showing that despite visual differences, DriveE2E reliably supports closed-loop, interaction-aware evaluation.
>
> # Response to Question 1:  Additional visualizations
> Due to the submission size constraints, we provide a limited number of driving scenario visualizations in the Supplementary Material.
>
> # Response to Question 2: CARLA blueprints and agent diversity
> We use CARLA’s blueprint library to render vehicles and VRUs, selecting assets to match agent categories and real-world physical dimensions. The strength of our digital twin lies in behavioral and interaction diversity, not visual variety. As shown in Table R1, DriveE2E captures denser urban intersections with more VRUs and complex multi-agent interactions than Bench2Drive.
>
> Agent instances are initialized according to the real-world trajectories recorded in each scenario, rather than random sampling, thereby preserving realistic traffic distributions and interaction sequences.
> |Benchmark|Vehicle|Cyclist|Pedestrian|
> |---------------|------|-----|-----|
> |Bench2Drive|19.88|0.04|0.12|
> |DriveE2E|28.81|2.54|2.19|
>
> Table R1. Average agents per scene. DriveE2E shows greater diversity and density of agents.
>
> # Response to Question 3: More illustration of real-world data collection
> We selected 15 intersections from the Beijing High-level Autonomous Driving Demonstration Zone (Figure 3a). At each intersection, four pairs of 25 Hz, 1920×1080 roadside cameras plus additional blind-spot cameras were mounted at elevated positions to ensure full coverage (Figure 3b) [1].
>
> **References**
> 1. Haibao Yu, et al. Dair-v2x: A large-scale dataset for vehicle-infrastructure cooperative 3d object detection. CVPR2022.
>
> # Response to Question 4: Explanation of element-wise similarity metrics
> We are happy to provide more details on the element-wise similarity metrics used in Tab. 7.  Different scene components are evaluated using metrics suited to their geometry:
>
> - Lane: IoU between CARLA and HD-map lane polygons, Hausdorff Distance for centerline deviations, and average relative lane-width error.
>
> - Crosswalk: IoU between CARLA and HD-map crosswalk polygons.
>
> - Buildings: IoU between CARLA (from OSM) and HD-map building footprints projected to the ground plane.
>
> - Road Mark & Plants: Verified via visual inspection against Google Earth imagery.
>
> - Agents (vehicles, cyclists, pedestrians): IoU between 3D bounding boxes in CARLA and the original dataset.
>
> These metrics ensure each element is evaluated in a manner best suited to its properties.

---

### Official Review · Reviewer_385z · 2025-10-20

**Soundness:** 2
**Presentation:** 3
**Contribution:** 2
**Rating:** 4
**Confidence:** 5

**Summary:**

This paper introduces DriveE2E, a closed-loop end-to-end autonomous driving benchmark that integrates real-world driving scenarios into the CARLA simulator. The authors extract  800 dynamic traffic scenarios from real world data collected by infrastructure-mounted sensors
and create digital twin assets for 15 real-world intersections. Then the authors analyze the data distribution and evaluate several state-of-the-art end-to-end autonomous driving methods on the benchmark.

**Strengths:**

1. The paper proposes a practical real-to-simulation workflow to bulid driving scenarios in CARLA simulator with the real world data.
2. The proposed infrastructure cooperation is effective to collect extra information for simulator.

**Weaknesses:**

1. The scale of the digital twin build by the benchmark is insufficient. Only containing 15 intersections, the digital twin is much smaller than the towns already in CARLA and the diversity of surroundings is limited. Additonally, the benchmark does not contain other road structures such as T-junctions or roundabouts.
2. The sim-to-real gap are not evaluated qualitatively. Adding figures or videos of scenarios in real world and simulator at the same location in the same view can show the gap clearly.
3. The simulation is only partly "closed-loop". The agents only replay the logs without interacting with the ego-vehicle, which hurts the realism of simulation.
4. How to make digital models of traffic agents is not mentioned. CARLA itself has a relatively small amount of vehicle models. Only selecting existing models for traffic agents in real world may lead to a large sim-to-real gap on the perception task of autonomous driving.

**Questions:**

1. Will the data processing code used to build the digital twins be open-sourced? These tools are helpful for the community to build more simulation scenarios with real world data.
2. It seems that the static intersection assets construction includes a lot of manual work. The high cost may make the digital twin construction unscalable. What is the typical time to build the digital twin of a single intersection manually? Are there any methods to reduce the manual construction time?

---

> ### Author Response · Authors · 2025-11-27
> **Rebuttal by Authors**
>
> We sincerely thank the reviewer for the insightful feedback. We respond to the comments point by point below.
> # Response to Weakness 1: Quality over Scale; Realism over Coverage.
> Our digital-twin maps are reconstructed directly from real-world traffic data; therefore, the current release includes 15 intersections corresponding to the surveillance videos we use. We emphasize that benchmark quality depends more on high-fidelity dynamic traffic data than on geographic size. In this regard, our benchmark provides richer and more realistic traffic dynamics than the CARLA Leaderboard.
>
> To address scene diversity, we have already built Y-junctions and roundabouts, which will be included in future releases.
>
> # Response to Weakness 2: Lack of Qualitative Evaluation of Sim-to-Real Gap
> We thank the reviewer for the suggestion. Currently, providing side-by-side qualitative comparisons is challenging because our real-world data are captured by **high-mounted infrastructure sensors**, while CARLA images come from **on-vehicle sensors**, resulting in different viewpoints. Direct alignment may not accurately reflect the scene.
>
> To complement this, we provide **quantitative metrics** for static environments, agent appearances, behaviors, traffic lights, and environmental conditions (Table 7), which objectively characterize sim-to-real fidelity.
>
> We plan to explore placing sensors at **corresponding roadside locations in CARLA** in the future, which could enable more accurate viewpoint-aligned visual comparisons.
>
> # Response to Weakness 3: Even partly "closed-loop", log-replay is core to our digital twins and creates a uniquely challenging setting
> We appreciate the reviewer’s observation regarding the partially closed-loop nature of our simulation, where non-ego agents replay logged trajectories without reacting to the ego vehicle, which indeed limits the realism of traffic interactions. We acknowledge this as a current limitation of the system.
>
> As briefly discussed in our **Response to Reviewer XQzd (Weakness 2)**, we adopted the log-replay protocol as a pragmatic choice due to the complexity and density of our urban intersection scenarios. In such highly congested settings with many traffic participants, even simple reactive controllers (e.g., IDM-based models) often lead to unstable or unrealistic behaviors, while more advanced interaction models introduce substantial computational overhead. Despite its simplicity, the log-replay setup creates a uniquely challenging evaluation environment in which surrounding agents never yield or adapt to the ego vehicle, forming a strictly non-cooperative scenario that forces the ego system to generate safe and feasible decisions beyond imitation of expert trajectories.
>
> Additionally, we plan to **augment the benchmark with interactive controllers** in future versions.
>
> **Finally, we emphasize that the log-replay mechanism is not merely a compromise but a key component of DriveE2E’s digital-twin methodology**. DriveE2E reconstructs 15 real-world intersections as static digital twins using high-resolution HD maps and incorporates precise trajectories derived from infrastructure sensors. Consequently, our scenarios are neither procedurally generated nor reconstructed solely from vehicle logs, but instead faithfully reproduce real-world road geometry, traffic signals, road features, and actual traffic flows—an essential factor in reducing the sim-to-real gap.
>
> # Response to Weakness 4:  More illustration of making digital models of traffic agents.
> We use the existing vehicle assets provided by CARLA and select those whose physical size closely match the real vehicles in our dataset. While we acknowledge that CARLA offers a limited variety of vehicle models, our benchmark focuses primarily on the end task of autonomous driving—planning—rather than perception. Therefore, as long as the simulated vehicles match the real ones in scale and kinematics, the impact on our target task remains limited.
>
> # Response to Question 1: Data Processing Code Release Plan
> We will release the data processing code used to build the digital twins and make it openly available for the community.
>
> # Response to Question 2: Human Workload of Current Digital Twin Pipeline
> Constructing the digital twin of a single intersection currently requires about 2 hours of manual work. To improve scalability, we are developing a pipeline that uses Google Earth–based visual recognition to automatically extract road geometry and environmental layout. Based on the detected dimensions, the system will be able to automatically generate corresponding CARLA roads and building assets, which will significantly reduce the amount of manual effort needed in future expansions.

---

### Official Review · Reviewer_o5n8 · 2025-10-28

**Soundness:** 2
**Presentation:** 2
**Contribution:** 2
**Rating:** 4
**Confidence:** 3

**Summary:**

This paper presents a real-to-sim digital-twin pipeline that ports real-world intersections into the CARLA simulator for closed-loop evaluation of autonomous driving systems. The authors reconstruct static environment assets (geometry/layout) to create digital twins of 15 real intersections, and use these scenes to evaluate multiple end-to-end driving baselines in fully closed-loop simulations. The goal is to enable repeatable, scenario-faithful testing in simulation while preserving key characteristics of real-world locations.

**Strengths:**

- `Well-motivated digital-twin evaluation: `Building digital twins for closed-loop assessment of end-to-end autonomous driving is timely and reasonable. The pipeline is clearly presented, and the multi-source asset creation—combining Blender tooling with OpenStreetMap, HD maps, and satellite imagery—is thoughtfully executed. The engineering effort is evident and appreciated.

- `Benchmarking value:` The paper provides extensive, apples-to-apples comparisons of popular end-to-end driving methods within the same digital-twin environments. This offers a strong reference baseline for the community and supports reproducibility and fair comparison.

**Weaknesses:**

- `Overlap with prior digital-twin platforms (ScenarioNet/MetaDrive):`
The core idea—building digital twins in a simulator for evaluation—appears closely related to prior work such as ScenarioNet/MetaDrive [1]. Please clarify the advantages over the previous work.

Reference:
[1] ScenarioNet: Open-Source Platform for Large-Scale Traffic Scenario Simulation and Modeling

- `Visual domain gap between real and simulated scenes:`
While digital twins enable controllable evaluation, the appearance gap between real images and simulated renderings (cf. Fig. 2) is noticeable and may bias conclusions for on-vehicle deployment. Please quantify and discuss the impact.

- `Unexpected baseline ranking (UniAD outperforming MomAD in Table 3):`
The result contrasts with MomAD’s strong closed-loop performance on Bench2Drive. In my opinion, there are indeed possible factors that include differences in sensor suites/resolution, control frequency/latency, action spaces and low-level controllers, training data/domain, metric definitions (e.g., Driving Score variants), and route/weather distributions. I am wondering the reason behind this difference from the authors' aspects.

**Questions:**

N/A

---

> ### Author Response · Authors · 2025-11-27
> **Rebuttal by Authors**
>
> # Response to Weakness 1:  Comparison with ScenarioNet  / MetaDrive
> Thank you for raising the concern regarding the potential overlap with *ScenarioNet / MetaDrive*. We fully understand this point and clarify that DriveE2E is complementary rather than redundant.
>
> Although ScenarioNet + MetaDrive and DriveE2E all follow the paradigm of “*digital twin + simulation + real-world traffic data*”, the systems differ fundamentally in motivation, fidelity, and application scope.
>
> **1. More Accurate Static and Dynamic Sim-to-Real Bridging**
>
> ScenarioNet reconstructs scenarios from vehicle logs (Waymo, nuScenes, L5, nuPlan) and standardizes them for replay inside MetaDrive. In contrast, DriveE2E rebuilds **15 real intersections** using **HD maps + infrastructure-sensor trajectories**, capturing accurate road geometry, traffic lights, and real multi-agent flows. No procedural generation is used. This provides better sim-to-real alignment for evaluating real-world feasibility of E2E driving.
>
> **2. Sensor-Level Differences**
>
> MetaDrive’s Panda3D renderer intentionally prioritizes efficiency, producing simplified camera/LiDAR outputs suitable for RL benchmarking but far from real-world distributions. DriveE2E relies on CARLA’s high-fidelity sensor simulation—multi-camera, multi-LiDAR, weather, lighting—combined with real intersection layouts and flows. This produces perceptual observations much closer to real-world data, which is crucial for closed-loop E2E evaluation.
>
>  **3. Closed-Loop End-to-End Evaluation Focus**
>
> DriveE2E focuses on evaluating end-to-end driving systems (perception → planning → control) in a closed-loop setting within realistic digital twins of real intersections.
>
> ScenarioNet / MetaDrive focus mainly on large-scale RL, scenario generation, and imitation learning.
>
> **Summary**: DriveE2E specializes in *real intersection reconstruction + real traffic flows + CARLA sensor simulation + closed-loop E2E evaluation*, making it fundamentally distinct and complementary to prior digital-twin platforms.
>
> # Response to Weakness 2:  Visual Domain Gap Between Real and Simulated Scenes
>
> We appreciate the reviewer’s concern regarding the appearance gap between CARLA-rendered imagery and real-world data. We acknowledge that CARLA is not fully photorealistic. To quantify the impact, we conducted a perception degradation study using BEVFormer (UniAD Stage-1)  trained separately on DriveE2E and nuScenes.
>
> The 3D object detection results in Table R1 show that when both training and evaluation use consistent sensor configurations, the “CARLA-train / CARLA-test” performance on DriveE2E is comparable to the “nuScenes-train / nuScenes-test” performance. This demonstrates that perception quality can remain stable in-domain, supporting the internal consistency of DriveE2E for end-to-end closed-loop evaluation.
>
> | Model                 | Dataset   | 3D Object Detection (Vehicle) |
> |-----------------------|-----------|--------------------------------|
> | BEVFormer-DriveE2E    | DriveE2E  | 0.632                          |
> | BEVFormer-nuScenes    | nuScenes  | 0.591                          |
>
> Table R1. 3D object detection results
>
> These observations align with the broader understanding that, for end-to-end autonomous driving, multi-agent behavioral realism is more critical than pixel-level photorealism, which is a conclusion also supported by prior E2E works that successfully evaluate CARLA-style simulated imagery.
>
> Therefore, while a visual domain gap remains, we believe its impact on driving strategy evaluation is limited, given that DriveE2E faithfully reconstructs real-world interaction patterns and provides consistent sensing conditions across training and testing. We plan to explore more advanced rendering pipelines in future releases.
>
> # Response to Weakness 3:  Unexpected Baseline Ranking (UniAD outperforming MomAD)
> The reviewer observes that UniAD performs better than MomAD on DriveE2E, opposite to results on Bench2Drive.
>
> **Firstly**, Bench2Drive and DriveE2E differ substantially in scenario characteristics. Bench2Drive uses simplified two-lane roads, rule-based agent behavior, and manually specified agent targets. In contrast, DriveE2E scenarios feature dense urban intersections and real traffic flows directly from infrastructure sensors.
>
> |Benchmark|Agent Number (vehicle)|Agent Number (cyclist)|Agent Number (pedestrian)|
> |-------------|-----------------------------|---------------------------|---------------------------------|
> |Bench2Drive|19.88|0.04|0.12|
> |DriveE2E|28.81|2.54|2.19|
>
> Table R2. Average agent quantities. DriveE2E has richer and more diverse agents.
>
> **Secondly**, We also observe that model performance varies across benchmarks: VAD performs better on Bench2Drive, whereas UniAD excels on DriveE2E. We have carefully re-verified MomAD integration and found no issues. We are still conducting a thorough investigation (For fairness, no closed-loop parameter tuning has been performed for any baseline).

---

### Official Review · Reviewer_XQzd · 2025-10-30

**Soundness:** 2
**Presentation:** 2
**Contribution:** 3
**Rating:** 4
**Confidence:** 5

**Summary:**

This paper presents DriveE2E, a Real2Sim-based closed-loop evaluation framework built upon the CARLA simulator. It features high-fidelity digital twins of 15 diverse urban intersections and 800 traffic scenarios generated from infrastructure sensor data.

**Strengths:**

The paper is well-written and clearly structured, making it easy to follow. It constructs high-fidelity digital twins of 15 urban intersections and selects 800 real-world traffic scenarios from over 100 hours of infrastructure sensor data. Meanwhile, it establishes a comprehensive closed-loop benchmark for end-to-end autonomous driving (E2EAD) by evaluating several classical baselines, demonstrating both technical completeness and practical relevance.

**Weaknesses:**

There are several points that require further clarification. The proposed Real2Sim framework is intended to address the unrealistic rendering issues in CARLA, yet the paper still relies on CARLA’s built-in assets for traffic participants, which seems inconsistent with that motivation. In addition, one of CARLA’s main advantages lies in its true closed-loop capability, where surrounding agents can dynamically respond to the ego vehicle’s behavior—unlike the log-replay mode adopted in this work. While this paper introduces a new scenario, it also employs a log replay method. Therefore, what are its advantages compared to frameworks like nuPlan or NAVSIM? This will determine the value of this work.

**Questions:**

Same to the Weaknesses.

---

> ### Author Response · Authors · 2025-11-27
> **Rebuttal by Authors**
>
> We thank the reviewer for the insightful comments. Below we address the concerns point-by-point.
> # Response to Weakness 1:  Why DriveE2E's Real2Sim pipeline still uses CARLA’s built-in traffic assets
> Our work targets scene-level realism rather than asset-level realism. The key limitation we aim to address is that existing CARLA environments lack real-world traffic **distribution, density, heterogeneity, spatial layout, and interaction patterns**. Even with photorealistic assets, these factors remain highly synthetic and do not reflect real-world dynamics.
> Real2Sim therefore focuses on reconstructing real intersections—including **road topology, lane geometry, traffic flows, trajectories, and interaction statistics**—and injecting them into CARLA’s closed-loop engine. Using CARLA’s native assets ensures **compatibility, controllability, and reproducibility**, and does not conflict with our motivation. Our contribution lies in bringing real-world behavioral realism and scenario realism to CARLA, not in redesigning 3D meshes of vehicles or pedestrians.
> # Response to Weakness 2: Why log-replay is chosen and what are its advantages compared to frameworks like nuPlan or NAVSIM
> **1. Why log-replay is chosen**
>
> In our current system, we adopt a **simplified closed-loop protocol—log-replay mode**— where all non-ego traffic participants strictly follow their recorded trajectories and do not respond to the ego vehicle’s actions. The main reason is that the DriveE2E scenes are concentrated in **dense urban intersections involving many traffic agents**. Introducing even simple rule-based reactive behaviors (e.g., IDM) often leads to **unstable or unrealistic behaviors** in such congested scenarios, while more advanced interaction models would impose **significant computational overhead** on the hardware.
>
> Therefore, we use **the simplest and most robust Real2Sim setup**: the ego vehicle is controlled by the E2EAD model, and all other agents directly replay their original logged trajectories. Despite its simplicity, this protocol actually creates a **more challenging driving environment**, because surrounding agents will never yield or adapt to the ego vehicle. The ego must therefore successfully drive in an environment that is **strictly non-cooperative**.
>
> Importantly, unlike open-loop evaluation, the ego vehicle in our closed-loop log-replay setting is **not required to imitate the human driver’s original trajectory**. The ego only needs to generate a safe and feasible driving strategy that reaches the destination under this adversarial environment. For example, an intersection may allow multiple valid paths. While the expert demonstration may choose to follow the preceding vehicle in one lane, it is equally acceptable for the ego vehicle to switch to a neighboring lane with fewer vehicles, as long as it maintains a **safe distance** from following traffic and avoids collisions. In such cases, the expert trajectory is **not the only valid solution**, and the ego can still complete the task successfully.
>
> **2. Comparison between DriveE2E and NAVSIM**
> - NAVSIM [1] is a non-reactive, open-loop evaluation benchmark that replays expert states at every timestep, uses short-horizon trajectory predictions to generate action sequences in a simulator, and scores feasibility based on collision-free execution. It does not provide closed-loop evaluation nor tests long-term policy robustness. In essence, NAVSIM still has not overcome the inherent shortcomings of open-loop models as highlighted in Table 6 of the paper.
> - In contrast, DriveE2E allows the ego vehicle to execute its own planned actions for the entire route without being constrained to the expert path. Novel-view observations are generated in real time by CARLA’s rendering engine, enabling arbitrary deviations from the demonstration route. This design—excluding the log-replay mode for non-ego agents—makes DriveE2E a truly fully closed-loop benchmark for end-to-end autonomous driving.
>
> [1] Dauner, D., Hallgarten, M., Li, T., Weng, X., Huang, Z., Yang, Z., ... & Chitta, K. (2024). Navsim: Data-driven non-reactive autonomous vehicle simulation and benchmarking. Advances in Neural Information Processing Systems, 37, 28706-28719.

---

### Meta-Review · Area_Chair_V7fg · 2026-01-07

**Summary:**

Reviewer concerns centered on the design choices of the Real2Sim framework, particularly the reliance on CARLA assets, the log-replay (non-reactive) setting for surrounding agents, limited scene scale and diversity, and the visual sim-to-real gap, as well as differentiation from existing benchmarks (ScenarioNet, MetaDrive, nuPlan, NAVSIM). Some concerns were addressed through clarification and justification. However, several issues remain partially addressed. Limitations related to non-interactive agents, restricted geographic coverage, manual construction cost, and the visual domain gap are acknowledged but remain inherent to the current release, with proposed improvements deferred to future work. As a result, while the rebuttal strengthened clarity and positioning, it does not majorly change the core limitations identified by reviewers, and reviewer scores are unlikely to increase. Overall, DriveE2E is viewed as a useful and well-motivated benchmark with clear practical value, but with scope and realism limitations that remain unresolved in the current version.

**Reviewer Concerns:**

Reviewer XQzd:

Addressed:
The reviewer’s concerns about the motivation and design choices of the Real2Sim framework were addressed through clarification that DriveE2E targets scene-level and behavioral realism rather than asset-level photorealism. The authors explained that using CARLA’s built-in traffic assets ensures compatibility and reproducibility, while realism is achieved by importing real intersection layouts, traffic flows, and interaction statistics from infrastructure data. The choice of log-replay was justified as a deliberate design trade-off to ensure stability and scalability in dense urban intersections, and its advantages over nuPlan and NAVSIM were discussed.

Partially Addressed:
While the authors acknowledged the limitations of non-reactive surrounding agents and positioned log-replay as a challenging but controlled closed-loop setting, fully interactive multi-agent behavior was not implemented in the current version. The lack of reactive traffic agents remains a limitation, though it is explicitly recognized and deferred to future work.

Reviewer o5n8:

Addressed:
The reviewer’s overlap concern with ScenarioNet/MetaDrive was addressed by explaining DriveE2E focuses on high-fidelity reconstruction of real intersections + real traffic flows using CARLA’s richer sensor simulation, whereas ScenarioNet/MetaDrive emphasize scalable scenario replay/RL with lighter-weight rendering.

Partially addressed:
The visual domain gap between real and simulated scenes was acknowledged and partially quantified through an in-domain perception study, showing stable performance when training and testing within DriveE2E. However, the appearance gap relative to real-world imagery remains, and its broader impact on cross-domain generalization was not fully resolved. The unexpected baseline ranking (UniAD > MomAD) was discussed and attributed to benchmark differences (denser, more diverse agents and real intersection flows in DriveE2E vs. more simplified Bench2Drive settings), but the authors note the root cause is still under investigation, so this point remains only partially resolved.

Reviewer 385z:

Addressed:
The reviewer’s questions regarding the construction pipeline transparency and reproducibility were addressed by the authors’ commitment to open-source the data processing tools used to build the digital twins, improving community usability and reproducibility.

Partially addressed:
Concerns about limited benchmark scale and scene diversity were acknowledged, with the authors clarifying their design choice to prioritize high-fidelity, data-driven intersection realism over geographic coverage. While the manual construction cost per intersection was quantified, scalability improvements rely on future automation plans, which are not yet realized.
Similarly, concerns regarding the sim-to-real visual gap, use of CARLA’s existing traffic assets, and the partially closed-loop (log-replay) interaction model were discussed and justified, but remain inherent limitations of the current benchmark. Qualitative visual comparisons and fully interactive agent behaviors are deferred to future work.

Reviewer 2DSC:

Addressed:
The reviewer’s requests for clarification on real-world data collection, agent diversity, and reconstruction fidelity metrics were addressed with detailed descriptions of the infrastructure sensor setup, agent categories and initialization, and element-wise similarity metrics used to evaluate digital-twin reconstruction quality. Additional statistics and explanations were provided to clarify the scope and realism of the benchmark.

Partially addressed:
Concerns regarding the log-replay simulation protocol, limited scene diversity and geographic coverage, and visual realism limitations of CARLA were acknowledged and justified. While the authors explained the rationale for log-replay in dense urban intersections and demonstrated that the benchmark supports reliable closed-loop evaluation, these limitations remain inherent to the current release. Extensions to interactive agent behaviors, broader intersection types, and improved visual diversity are planned for future versions but are not yet realized.

**Reviewer Scores:**

Reviewer XQzd would possibly maintain their score of 4.
Reviewer o5n8 would possibly maintain their score of 4.
Reviewer 385z would possibly maintain their score of 4.
Reviewer 2DSC would possibly maintain their score of 4.

---

### Decision · Program_Chairs · 2026-01-26

Reject